# Robustness through Data Augmentation Loss Consistency

**Tianjian Huang**[*]                                                                        *tianjian@usc.edu*
*University of Southern California*

**Shaunak Halbe**[*][†]                                                                      *shalbe9@gatech.edu*
*Georgia Institute of Technology*

**Chinnadhurai Sankar**                                                                     *chinnadhurai@meta.com*
*Meta AI*

**Pooyan Amini**                                                                            *pamini@meta.com*
*Meta AI*

**Satwik Kottur**                                                                           *skottur@meta.com*
*Meta AI*

**Alborz Geramifard**                                                                       *alborzg@meta.com*
*Meta AI*

**Meisam Razaviyayn**[*]                                                                     *razaviya@usc.edu*
*University of Southern California*

**Ahmad Beirami**[†]                                                                         *beirami@google.com*
*Google Research*

**Reviewed on OpenReview:** *https://openreview.net/forum?id=a1meaRy1bN*

## Abstract

While deep learning through empirical risk minimization (ERM) has succeeded at achieving human-level performance at a variety of complex tasks, ERM is not robust to distribution shifts or adversarial attacks. Synthetic data augmentation followed by empirical risk minimization (DA-ERM) is a simple and widely used solution to improve robustness in ERM. In addition, consistency regularization can be applied to further improve the robustness of the model by forcing the representation of the original sample and the augmented one to be similar. However, existing consistency regularization methods are not applicable to *covariant data augmentation*, where the label in the augmented sample is dependent on the augmentation function. For example, dialog state covaries with named entity when we augment data with a new named entity. In this paper, we propose data augmented loss invariant regularization (DAIR), a simple form of consistency regularization that is applied directly at the loss level rather than intermediate features, making it widely applicable to both invariant and covariant data augmentation regardless of network architecture, problem setup, and task. We apply DAIR to real-world learning problems involving covariant data augmentation: robust neural task-oriented dialog state tracking and robust visual question answering. We also apply DAIR to tasks involving invariant data augmentation: robust regression, robust classification against adversarial attacks, and robust ImageNet classification under distribution shift. Our experiments show that DAIR consistently outperforms ERM and DA-ERM with little marginal computational cost and sets new state-of-the-art results in several benchmarks involving covariant data augmentation. Our code of all experiments are available at: `https://github.com/optimization-for-data-driven-science/DAIR`.

---

[*]The work of TH, SH, and MR was partially supported with funding from the USC-Meta Center for Research and Education in AI and Learning (REAL@USC) and with gift funding from 3M.

[†]The work of SH was done at the University of Southern California and the work of AB was done at Meta AI.

# 1 Introduction

Deep neural networks are widely used in various applications ranging from computer vision to language processing. While deep learning has surpassed human-level performance in numerous tasks, neural networks fail under small adversarial perturbations of the test samples (Goodfellow et al., 2015) or natural shifts of distribution at deployment time (Arjovsky et al., 2019). These issues have motivated the research community to invest in a variety of methods for evaluation and mitigation of *robustness* in deep learning. This includes introduction of new robustness benchmarks, e.g., Rotated/Colored MNIST (Arjovsky et al., 2019), DomainNet (Peng et al., 2019), ImageNet-R (Hendrycks et al., 2021), ImageNet-9 (Xiao et al., 2020) for robustness to natural distribution shift; and Fast Gradient Sign Method (FGSM) (Goodfellow et al., 2014) and Projected Gradient Descent (PGD) (Madry et al., 2018) for adversarial robustness.

Researchers have also proposed numerous algorithmic solutions to improve robustness to distribution shift (Ganin et al., 2016; Ghifary et al., 2015; Sagawa et al., 2019; Li et al., 2018a; Sun & Saenko, 2016; Li et al., 2018b;c; Krueger et al., 2021; Zhang et al., 2021; Robey et al., 2022) and adversarial attacks (Madry et al., 2018; Li et al., 2020; Zheng et al., 2020; Zhang et al., 2019; Tack et al., 2021). These approaches are usually more complex than conventional empirical risk minimization (ERM) and hence they cannot be readily applied to involved tasks with non-trivial model architectures. For example, in generative language modeling imposing a constraint on the intermediate data representations is non-trivial, which is required by CORAL (Sun & Saenko, 2016). In addition, recently Gulrajani & Lopez-Paz (2020) demonstrated that vanilla ERM, if tuned properly, remains competitive with (or may even outperform) many such complex methods in real-world scenarios involving robust inference.

**Data augmentation** can be employed to improve the robustness of ERM by curating synthetic examples that exhibit a desired invariance/covariance. In this paper, *invariant data augmentation* refers to the case where the features are perturbed to obtain a synthetic augmented example that preserves the original label. For example, in an image classification task where the aim is to classify the foreground of an image, we can create augmented data points by keeping the foreground and changing the background. Under this data augmentation process, the label of data (i.e., the image class) does not change and hence we call it invariant data augmentation. On the other hand, *covariant data augmentation* refers to the case where perturbation of the features results in the label to covary with the features. For example, in a dialog state tracking task where data augmentation is performed by varying named entities in the input data, the model output (label) of the augmented data also needs to covary to contain the new named entity.

Data augmentation techniques abound in the literature: Tensmeyer & Martinez (2016) and Cutout (DeVries & Taylor, 2017) curate invariant image transformations to improve image representations. Mixup (Zhang et al., 2017) and CutMix (Yun et al., 2019) curate covariant data augmentations via linear combination of features between different classes to learn more robust representations. Volpi et al. (2018); Zhou et al. (2020) perform data augmentation with adversarial images to improve robustness. Finally, Cubuk et al. (2018); Lim et al. (2019) introduce a procedure which automatically searches for improved data augmentation policies. While simple, data augmentation remains an effective and universal solution to improve model robustness.

**Consistency regularization** can be further applied on top of data augmentation to enhance robustness by enforcing the desired invariances on the model. Engstrom et al. (2018); Kannan et al. (2018); Zhang et al. (2019); Tack et al. (2021) utilize consistency regularization at an embedding layer to train robust neural networks against adversarial attacks. Various forms of consistency regularization have been applied to unsupervised learning (Sinha & Dieng, 2021), self-supervised learning (Chen et al., 2020; von Kügelgen et al., 2021), and semi-supervised learning to exploit unlabeled data (Bachman et al., 2014; Laine & Aila, 2016; Miyato et al., 2018; Sohn et al., 2020; Xie et al., 2020). Standard consistency regularization forces intermediate features to be similar among all inputs variations and hence is only applicable to invariant data augmentation, where data augmentation keeps the label of the augmented sample intact. Such consistency regularization may even hurt performance in the face of covariant data augmentation, where the label for the augmented sample may change. For example, in dialog state tracking where the dialog state covaries with named entities in dialog context, standard consistency regularization hurts overall performance as it forces the model to not depend on the named entity for its prediction. See Section 2.1 for a more detailed explanation and Section 3 for experiments that confirm this.

In this paper, we propose a simple form of consistency regularization, called data augmented loss invariant regularization (DAIR), that is directly applied at the loss level. While existing consistency regularization methods can only be applied to invariant data augmentation, DAIR is applicable to both invariant/covariant data augmentation when a pair of data samples expecting consistent performance. This is vastly in contrast to existing feature consistency regularizers that apply on an intermediate embedding space or the very last layer of the model (logits). As a result, DAIR only requires marginal computational cost on top of data augmentation, and is broadly applicable to a wide host of learning tasks, including generative models with covariant data augmentation. We theoretically prove some of the properties of DAIR in Section 2 and empirically evaluate DAIR on a variety of problems ranging from defense against adversarial attacks to domain generalization in the presence of environment shift in Sections 3 and 4. Our experimental results show that DAIR is competitive with state-of-the-art algorithms specifically designed for these problems.

## 2 DAIR: Data Augmented Loss Invariant Regularization

For a data sample $z = (x, y)$, let $\ell(z; \theta)$ be its parametric loss function, where $\theta$ is the set of model parameters (e.g., network weights). The popular Empirical Risk Minimization (ERM) framework trains the model by minimizing the expected value of the following loss over the training data:

$$f_{\text{ERM}}(z; \theta) = \ell(z; \theta). \tag{ERM}$$

We assume that we have access to a (potentially randomized) data augmenter function $A(\cdot)$. Examples for $A$ include (random) rotation, change of background, or change of entity names. Such augmenters capture the transformations against which we wish to be invariant. Given a sample $z$, let $\widetilde{z} = (\widetilde{x}, \widetilde{y}) = A(z)$ denote an augmented sample. Previous work has used both original and augmented examples during training, which leads to the following standard objective function, called Data Augmented Empirical Risk Minimization (DA-ERM):

$$f_{\text{DA-ERM}}(z, \widetilde{z}; \theta) = \frac{1}{2}\ell(z; \theta) + \frac{1}{2}\ell(\widetilde{z}; \theta). \tag{DA-ERM}$$

While DA-ERM has been successful in many applications, one natural question is whether we can further improve upon it using the knowledge that the performance on augmented samples should be consistent with the original ones. Consistency regularization further penalizes DA-ERM for any such inconsistency at the feature/loss level: $f_{\text{Consistency}, \mathcal{D}, \lambda}(z, \widetilde{z}; \theta) = f_{\text{DA-ERM}}(z, \widetilde{z}; \theta) + \lambda \mathcal{D}(z, \widetilde{z}; \theta)$, where $\mathcal{D}(z, \widetilde{z}; \theta)$ is a proper divergence between the original sample representation and the augmented sample representation, and where the goal of the regularizer applied at some intermediate feature space is to maintain the performance of the model on $z$ and $\tilde{z}$ consistent. In this paper, we focus on a specific type of such regularization, called data augmented loss invariant regularization (DAIR):[*]

$$f_{\text{DAIR}, \mathcal{R}, \lambda}(z, \widetilde{z}; \theta) = f_{\text{DA-ERM}}(z, \widetilde{z}; \theta) + \lambda \mathcal{R}(\ell(z; \theta), \ell(\widetilde{z}; \theta))$$
$$= \frac{1}{2}\ell(z; \theta) + \frac{1}{2}\ell(\widetilde{z}; \theta) + \lambda \mathcal{R}(\ell(z; \theta), \ell(\widetilde{z}; \theta)), \tag{DAIR}$$

where the regularization is directly applied to the loss. This "loss-level" regularization approach is different from existing "feature-level" consistency regularization approaches discussed in section 1. For example, consider a classification task with a neural network $\mathcal{F}(\cdot, \theta) : \mathbb{R}^d \mapsto \mathbb{R}^k$ where $\theta$ denotes the weights of the network. This network takes a $d$-dimensional input data point and outputs a $k$-dimensional logit/softmax output. A popular training approach is to consider $\ell(z; \theta) = C(\mathcal{F}(x, \theta), y)$ where $C(\cdot, \cdot)$ is the cross-entropy loss and $z = (x, y)$. Existing regularization mechanisms, such as Engstrom et al. (2018); Kannan et al. (2018); Zhang et al. (2019); Tack et al. (2021); Sinha & Dieng (2021); von Kügelgen et al. (2021); Miyato et al. (2018), to name just a few, operate at the feature level, i.e., they are of the form $\mathcal{R}_f(\mathcal{F}(x, \theta), \mathcal{F}(\widetilde{x}, \theta))$. However, DAIR operates at the loss level as demonstrated in eq. (DAIR). The idea behind DAIR is to simply promote $\ell(z; \theta) \approx \ell(\widetilde{z}; \theta)$, and ignore the features or even the rest of the possible outcomes of $y$ and simply focus on the

---

[*]We performed our initial experiments by weighting the original and augmented examples differently, and empirically observed that the weight had no significant effect on the performance. Hence, we fixed equal weight on both original and augmented samples for ease of tuning.

current sample's loss. While DAIR is a relatively weaker form of consistency regularization, it is suitable for problems where feature consistency may not be conceptually meaningful (See Section 2.1 for a more detailed discussion). For instance, in language modeling when a pair of sentences differ in their corresponding named entities, it is not clear why we should enforce their embeddings to be similar unless the label is desired not to depend on such named entity. However, loss consistency is still meaningful promoting the probability of label given input to be the same on the original and the augmented samples.

We remark that DAIR requires pairing information between original and augmented samples, which may not always be available (e.g., DomainBed (Gulrajani & Lopez-Paz, 2020)). However, we show that this simple approach is still broadly applicable to various real-world problems regardless of model architecture, and is indeed competitive with state-of-the-art methods for imposing invariance. As it turns out, we are particularly interested in a particular form of the DAIR regularizer:

$$\mathcal{R}_{\mathrm{sq}}(z, \widetilde{z}; \theta) := \left( \sqrt{\ell(z; \theta)} - \sqrt{\ell(\widetilde{z}; \theta)} \right)^2, \tag{DAIR-SQ}$$

and we call this variant DAIR-SQ. Note that $\mathcal{R}_{\mathrm{sq}}$ has the same scale as the loss function $\ell$, making it easier to tune $\lambda$. Empirically we observe that the optimal $\lambda$ for all the experiments mentioned later in the paper falls in $[0.2, 100]$, across various tasks (from regression to sequence-to-sequence generative modeling). Further justification on DAIR-SQ will be provided through the rest of this section.

Finally, in most (real-world) applications performance is measured through 0-1 metrics other than the loss function. For example, we are usually concerned with accuracy in image classification while we optimize cross-entropy loss. Let $\mathcal{H}(z; \theta) \in \{0, 1\}$ denote a 0-1 evaluation performance metric of interest, e.g., accuracy. Given the sample $z$ (or $\widetilde{z}$), the model performance is captured by $\mathcal{H}(z; \theta)$ (or $\mathcal{H}(\widetilde{z}; \theta)$). For any $z$ such that $\mathcal{H}(z; \theta) = 1$, we define the corresponding consistency metric as:

$$\mathrm{CM}(z, \widetilde{z}; \theta) = \mathbb{I}\{\mathcal{H}(\widetilde{z}; \theta) = 1 \mid \mathcal{H}(z; \theta) = 1\}. \tag{Consistency Metric}$$

Notice that similarly to the original performance metric, which is only used for model evaluation, we use the consistency metric at evaluation time only.

## 2.1 Why DAIR at the loss level?

As we mentioned in Section 1, consistency regularization has been extensively studied in the literature. However, regularization at loss level has been relatively unexplored. We propose DAIR at the loss level, making it broadly applicable when pairing information is available between original and augmented samples even under the cases where the respective labels are different. Consider the following two examples: visual question answering (Section 3.2) and dialog state tracking (Section 3.1) in which the labels of the augmented examples covary with the augmented features. In these setups, feature consistency regularization is not conceptually meaningful as the embedding of the image with zebra removed should not be the same as the original image (Figure 1), or the embedding of the dialog state with named entity changed from airport to bus station should not remain unchanged (Figure 2). In fact, forcing the embeddings to be exactly the same will remove vital information needed for performing the task and will incorrectly force the model to provide the same output for the original and augmented samples. On the other hand, we can enforce the loss value at the augmented sample and the original sample to be the same, which implies $p(\widetilde{y}|\widetilde{x}; \theta) \approx p(y|x; \theta)$ when loss is viewed as a log-likelihood function.

To contextualize DAIR, consider a classification task using a function approximator (e.g., a deep neural network) followed by a softmax layer. Let $\mathcal{F}(x, \theta)$ be the output of the model right before the softmax layer. Hence, $\mathcal{F}(x, \theta) \propto e^{-\ell(x, \cdot; \theta)}$ for all possible outcomes In addition to two DAIR variants, we consider the regularizer to be any proper divergence between the output distributions $\mathcal{F}(x, \theta)$ and $\mathcal{F}(\widetilde{x}, \theta)$, such as KL divergence, which will promote $\mathcal{F}(x, \theta) \approx \mathcal{F}(\widetilde{x}, \theta)$. In addition to DAIR-SQ, we define the following regularizers that we use throughout the paper:

Q: How many zebras are there in the picture?
A: 2        *zebra removed* A: 1

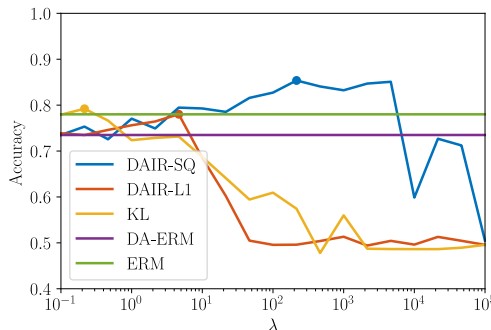

Figure 1: VQA: Answer changes after augmentation. Image taken from Agarwal et al. (2020).

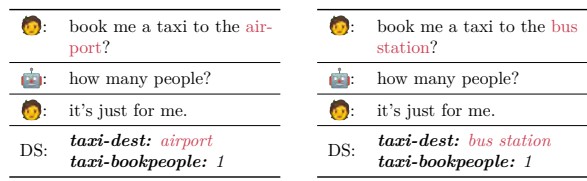

Figure 2: DST: dialog state changes after augmentation.

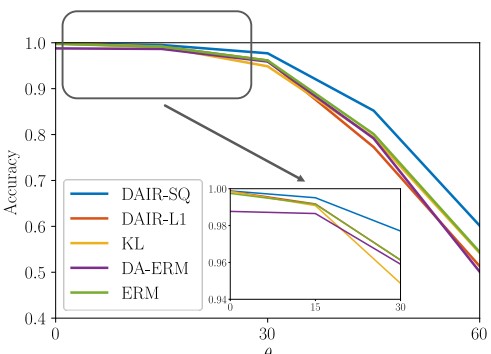

Figure 3: Test accuracy for $\theta = 45°$ rotated images as a function of $\lambda$ (strength of the regularizer). Best validation $\lambda$ is marked with solid circles.

Figure 4: Test accuracy as a function of test-time rotation $\theta \in [0°, 60°]$. We report the performance of each method with their best validation $\lambda$.

- $\mathcal{R}_{\mathrm{L1}}(z, \widetilde{z}; \theta) := |\ell(z; \theta) - \ell(\widetilde{z}; \theta)|;^{\dagger}$       (DAIR-L1)
- $\mathcal{R}_{\mathrm{KL}}^{\mathcal{F}}(z, \widetilde{z}; \theta) := \mathrm{KL}(\mathcal{F}(x, \theta) \| \mathcal{F}(\widetilde{x}, \theta)).$       (KL Feature Consistency)

Notice that the KL feature consistency regularizer is oblivious to $\widetilde{y}$, and remains the same even for covariant data augmentation where $\widetilde{y} \neq y$.

**Toy Experiment: Rotated 6 & 9.** Consider a binary classification problem which is derived from MNIST with only digits 6 and 9. The goal is to explore test-time distribution shifts in the form of rotation. We consider the augmentation function to be 180° rotations. Since the digits 6 and 9 are similar to each other respectively after such rotation, we naturally flip the augmented label, making this a covariant data augmentation. The rotation schemes for training, augmentation and testing are detailed in Table 1.

In Figure 3, we benchmark the performance of KL feature consistency, DAIR-L1, and DAIR-SQ against ERM and DA-ERM, when the test rotation is $\theta = 45°$, as a function of regularizer strength. As can be seen, DAIR-SQ (blue) is the only approach that outperforms the ERM baseline (green). This is not a coincidence. Our 180° rotation augmentation is covariant and it can be seen that vanilla DA-ERM does not help but even hurts the performance, which may be attributed to the fact that the test rotations are novel and unseen. KL feature consistency (yellow) slightly outperforms DA-ERM for small $\lambda$ but struggles to outperform ERM. This is a covariant data augmentation case, and hence feature consistency was expected to fail. We see the performance of KL degrades as $\lambda$ gets larger. It is noteworthy that DAIR-L1 (red) starts to degenerate to a random classifier baseline for large values of $\lambda$ and fails to produce a model that outperforms ERM. We will investigate this issue more deeply in the next section.

To further explore how the performance of these baselines are affected by the degree of test-time distribution shift, we consider a variety of test sets with the degree of rotation $\theta$ swept in $[0°, 60°]$. In Figure 4, we report

---

$^{\dagger}$A similar L1 regularization idea has been explored in (Garg et al., 2019), however, our proposed DAIR regularizer differs substantially by imposing consistency at the loss level.

| Split | Rotation | Example Input | Example Label |
|-------|----------|---------------|---------------|
| Train | $0°$ | 6 | 6 |
| Augmentation | $180°$ | 9 | 9 |
| Test | $\theta \in [0°, 60°]$ | 6 | 6 |

Table 1: The rotation scheme used in the Rotated 6 & 9 experiment.

the performance of each algorithm for different values of $\theta$. We see that DA-ERM hurts in-distribution accuracy ($0°$ rotation), and generally hurts test accuracy for various degrees of rotation. For relatively large $\theta$ (representing relatively large distribution shift), only DAIR-SQ consistently outperforms the ERM baseline. Further, since the data augmentation scheme was weak and did not match the test-time distribution, it was expected that vanilla data augmentation is not sufficient to help the model to generalize to the test-time shift. It is noteworthy that perhaps surprising DAIR-SQ outperforms ERM (by a large margin for big distribution shifts) even though DA-ERM is underperforming ERM.

Through this toy experiment, we motivated the significance of loss-level regularization through DAIR-SQ. To theoretically analyze why/how DAIR-SQ works, we also conduct a simple toy linear regression experiment in Appendix A followed by a multi-dimensional extension in Appendix B, where we provide formal proofs to show that DAIR-SQ is guaranteed to outperform DA-ERM, even in the regime of infinite data or when using weight decay regularization.

## 2.2 Practical considerations for DAIR

Next, we discuss a more detailed comparison of DAIR-SQ with DAIR-L1, and its dependency on $\lambda$, where we explain the rationale for settling on DAIR-SQ for the experiments in Section 3.

### 2.2.1 Why does DAIR-SQ significantly outperform DAIR-L1?

While we have already compared DAIR-SQ with several consistency regularization alternatives, we want to specifically focus on a closely related DAIR variant called DAIR-L1, which has already appeared in the literature for invariant data augmentation (Garg et al., 2019). As we observed in Section 2.1, DAIR-L1 either outright failed or was unstable on the toy example. The following lemma further investigates the discrepancy between DAIR-SQ and DAIR-L1:

**Lemma 1.** *For any non-negative loss function $\ell$,*

$$\mathcal{R}_{L1}(z, \widetilde{z}; \theta) - \mathcal{R}_{sq}(z, \widetilde{z}; \theta) \geq 0,$$

*with equality iff* $\min\{\ell(\widetilde{z}; \theta), \ell(z; \theta), \ell(\widetilde{z}; \theta) - \ell(z; \theta)\} = 0.$

The proof of Lemma 1 appears in Appendix C.1. The difference is depicted in Figure 5. This suggests that $\mathcal{R}_{\mathrm{sq}}(z, \widetilde{z}; \theta)$ incurs a much smaller penalty when $\ell(z; \theta)$ is large. On the other hand, when $\ell(z; \theta) \approx 0$ the regularizer is much stronger and almost equivalent to $\mathcal{R}_{\mathrm{L1}}$. Why does this matter? At the beginning of training when the network is not yet trained, the loss values on the original samples are large, and $\mathcal{R}_{\mathrm{sq}}$ regularizer is weak letting the training to proceed towards a good solution for the original samples. As the network is being trained on original samples and their loss is vanishing, the regulairzer starts to force the network to become invariant on the augmented samples. The above hypothesis is also empirically verified on Colored MNIST with Adversarial Augmentation (see Appendix F.3).

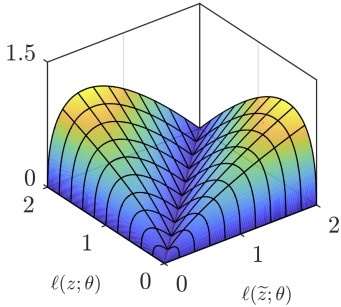

Figure 5: The plot of $\mathcal{R}_{\mathrm{L1}}(z, \widetilde{z}; \theta) - \mathcal{R}_{\mathrm{sq}}(z, \widetilde{z}; \theta)$.

We also explore the impact of partial augmentation, where we only augment a certain fraction of the training samples. DAIR shows stable performance compared with DA-ERM when the number of augmented examples are limited. The results are presented in Figure 18 (Appendix G).

### 2.2.2 Dependence of DAIR-SQ on the regularization strength $\lambda$

While we have already seen that practically the optimal $\lambda$ lies in the range $[0.2, 100]$, in this section we relate $\lambda$ to the quality of the solution.

**Definition 2** (Empirical Expectation). *We use $\widehat{\mathbb{E}}$ to denote the empirical expectation over a set of examples $\mathcal{S}$.*

$$\widehat{\mathbb{E}}_z \, \mathcal{L}(z) = \frac{1}{|\mathcal{S}|} \sum_{i \in \mathcal{S}} \mathcal{L}(z_i).$$

**Proposition 3.** *Let $\theta_\lambda^\star \in \arg\min_\theta f_{DAIR,\mathcal{R}_{sq},\lambda}(z, \widetilde{z}; \theta)$ and $\widetilde{\theta}$ denote any perfectly invariant solution, i.e., $\mathcal{R}(\ell(z; \widetilde{\theta}), \ell(\widetilde{z}; \widetilde{\theta})) = 0$. We have:*

$$\widehat{\mathbb{E}}_z \left\{ \mathcal{R}_{sq}(\ell(z; \theta_\lambda^\star), \ell(\widetilde{z}; \theta_\lambda^\star)) \right\} \leq \widehat{\mathbb{E}}_z \left\{ \frac{\ell(z; \widetilde{\theta}) + \ell(\widetilde{z}; \widetilde{\theta})}{2\lambda} \right\}.$$

Proposition 3 bounds the value of equation DAIR-SQ inversely proportional to $\lambda$. Consider the example of a classification task with $K$ classes where the number of samples in different classes are the same. When the weights are zero, i.e., $\widetilde{\theta} = \mathbf{0}$, we have a perfectly invariant solution. Moreover, for this choice, we have $\ell(z; \widetilde{\theta}) = \log(K)$ for all $z$ if cross entropy loss is used. The above lemma implies that $\widehat{\mathbb{E}}_z \{\mathcal{R}_{sq}(\ell(z; \theta_\lambda^\star), \ell(\widetilde{z}; \theta_\lambda^\star))\} \leq \frac{\log K}{\lambda}$. In other words, we can impose invariance by increasing $\lambda$ but we don't need a very large $\lambda$, reconfirming that $\lambda \leq 100$ sufficed in all of our experiments.

Although we have shown above that $\lambda$ needs not to be very large, a natural question is that could we choose a large $\lambda$ anyway since when $\lambda \to \infty$, the resulting model is perfectly invariant. In the following theorem, we show DAIR-SQ leads to convergent algorithms when optimized by popular methods such as gradient descent and the convergence rate is affected by $\lambda$. We remark that the proof of Proposition 3 can be extended to cover the setup of DAIR-L1 as well.

**Theorem 4.** *Consider a classification problem with logistic loss, where $\mathbf{x}$, $\widetilde{\mathbf{x}}$ is the input, $y$ is the output and $\theta$ denotes model parameters. Assume $\|\mathbf{x}\|, \|\widetilde{\mathbf{x}}\| \leq \mathcal{D}_x$, and $\|\theta\| \leq \mathcal{D}_\theta$. After $T$ iterations of gradient descent algorithm (Algorithm 1 in Appendix C.3), we have*

$$\|\nabla_\theta f_{DAIR,\mathcal{R}_{sq},\lambda}(\cdot)\|_2 = \mathcal{O}\left( \frac{(1 + \lambda\sqrt{\mathcal{D}_x \mathcal{D}_\theta})\mathcal{D}_x^2}{\sqrt{T}} \right).$$

Theorem 4 shows that DAIR-SQ penalizes the convergence rate for solving the problem to $\epsilon$-gradient accuracy by a $\lambda\sqrt{\mathcal{D}_x \mathcal{D}_\theta}$ factor as $\lambda$ increases. The penalty grows linearly with $\lambda$, which is consistent with the observation in Appendix F.2. However, we observe that the additional complexity is negligible in practice as we usually do not solve the optimization problems to stationarity but rather stop after certain number of iterations. For all experiments, we solve each task for a certain number of epochs, which is chosen the same as the ERM baseline. While Theorem 4 is established for the gradient descent, it can be extended to the stochastic settings (Lei et al., 2019, Theorem 2) based on the smoothness of the DAIR-SQ loss established in the proof of Theorem 4.

## 3 Experiments on covariant tasks

Thus far, we observed that DAIR-SQ is a practically stable variant of DAIR with some theoretical motivations and guarantees (such as convergence). In the rest of the paper, when we refer to DAIR without only postfix, we mean DAIR-SQ. As we empirically evaluate the performance of DAIR, we emphasize that the only hyperparameter that we tune for DAIR is $\lambda$ (chosen via grid search on validation set). The rest of the hyperparameters, such as step-size, batch-size, and number of training epochs, are only tuned for the ERM baseline and chosen to be exactly the same for DAIR. In this section, we continue with covariant tasks where feature-level regularization is expected to hurt the performance.

### 3.1 Neural task-oriented dialog modeling

Virtual digital assistants that engage in conversations with human users are rapidly gaining popularity. These devices require the modeling of task-oriented dialog systems that can communicate with users through natural language to accomplish a wide range of tasks. One of the main objectives in task-oriented dialog systems is the Dialog State Tracking (DST), which refers to keeping track of the user goals as the conversation progresses. Among task-oriented dialog datasets, MultiWOZ (Budzianowski et al., 2018) has gained the most popularity owing to the availability of 10k+ realistic dialogs across 8 different domains, and has been improved several times (Wu et al., 2019; Eric et al., 2019; Zang et al., 2020; Han et al., 2021; Qian et al., 2021).

Recently, SimpleTOD (Hosseini-Asl et al., 2020) achieved state-of-the-art results on MultiWOZ using a neural end-to-end modeling approach. However, Qian et al. (2021) observed that the performance of SimpleTOD drops significantly when the test set named entities (which are places in the UK) are replaced with new ones never observed during training (with new entities all based in the US), perhaps due to the memorization of named entities during training. We leverage DAIR to promote invariance of the dialog policy to named entities in the dialog flow. More importantly, we show that standard consistency regularization on feature space simply does not work. Here, the data augmentation scheme is a simple one. We replace named entities in the training set with their randomly scrambled version. For example, "cambridge" could be turned into "bmcedrgia." Details on training data, augmentation schemes and hyper-parameters can be found in Appendix H.

|  | MultiWOZ 2.2 Test JGA | MultiWOZ 2.2 Test JGA w/ SGD entities | CM |
|---|---|---|---|
| SimpleTOD (Hosseini-Asl et al., 2020) | 0.5483 | 0.4844 | 0.8206 |
| SimpleTOD + DA | 0.5915 | 0.5311 | 0.8354 |
| SimpleTOD + KL feature consistency | 0.5124 | 0.4053 | 0.8298 |
| SimpleTOD + DAIR | **0.5998** | **0.5609** | **0.8902** |

Table 2: DAIR achieves state-of-the-art Joint Goal Accuracy (JGA) on both the original MultiWOZ 2.2 test set (Zang et al., 2020) and well as the MultiWOZ 2.2 test set w/ named entities replaced with SGD (Qian et al., 2021).

The results are presented in Table 2, where performance is measured in Joint Goal Accuracy (JGA). JGA is a binary metric, and is equal to 1 if the predictions of all dialog states in a turn are correct. As such it is a difficult metric to get right too. As can be seen, both DA-ERM and DAIR outperform SimpleTOD (Hosseini-Asl et al., 2020) on MultiWOZ 2.2 w/ SGD entities (Qian et al., 2021). More surprisingly, DAIR also outperforms SimpleTOD on the original MultiWOZ 2.2 test set with no distribution shift, which we attribute to better robustness to the named entity memorization problem observed by Qian et al. (2021); Cho et al. (2022). We also observe that DAIR significantly improves the JGA consistency metric compared to the DA-ERM baseline. It worth nothing but the simple regularizer provides non-trivial performance increase which takes researchers several years of explore. Finally, we show that standard consistency regularization (KL) is not well suited to this task and results in performance degradation (see Section 2.1 for more explanation on why).

### 3.2 Covariant Visual Question Answering

Visual Question Answering (VQA) has diverse applications ranging from visual chatbots to assistants for the visually impaired. In such real-world settings, it is desirable for VQA models to be robust to variations in the input modalities. In this spirit, recent works (Agarwal et al., 2020; Shah et al., 2019; Ray et al., 2019) have studied the robustness and consistency of VQA models under linguistic and visual variations. In this paper, we focus on the Invariant and Covariant VQA (IV/CV-VQA) dataset which contains semantically edited images corresponding to a subset from VQA v2 (Goyal et al., 2017). CV-VQA contains images constructed by removing an object which is relevant to answering the question and leads to a different answer. A robust model should make predictions that are consistent to such edits.

We choose the attention based SAAA (Kazemi & Elqursh, 2017) model to match the original setup from Agarwal et al. (2020). Using DAIR, we enforce consistency in predictions between the original and edited samples. Wherever the edited image is not available, the DAIR formulation reduces to ERM. We use the

standard VQA accuracy along with our consistency metric to compare our results against the ERM and DA-ERM setups discussed in Agarwal et al. (2020). In addition we compare against a KL feature regularizer to highlight the strengths of our approach.

| Algorithm | CV-VQA test ↑ | CM ↑ |
|---|---|---|
| ERM (Kazemi & Elqursh, 2017) | 45.89 | 0.5792 |
| DA-ERM (Agarwal et al., 2020) | 48.32 | 0.5631 |
| KL Feature Consistency | 48.20 | 0.3479 |
| DAIR | **49.75** | **0.7161** |

Table 3: Accuracy and Consistency metrics on CV-VQA test set

The results for the CV-VQA are in Table 3. DAIR achieves a higher accuracy as compared to all baselines. This improvement is significant given that the model needs to predict the answer correctly from 3000 candidate answers. As against this, applying KL for feature consistency catastrophically fails on the CV-VQA task achieving significantly lower CM scores than ERM. We describe the Invariant VQA experiment is Section 4.3.

## 4 Experiments on invariant tasks

Now that we have established the effectiveness of DAIR on covariant tasks, we also benchmark its performance on invariant tasks where more baselines are available. As in the previouss section, we only tune for $\lambda$ via grid search and all other hyperparameters remain the same as the ERM baseline.

### 4.1 Colored MNIST

Colored MNIST (Arjovsky et al., 2019) is a binary classification task built on the MNSIT dataset. Digits 0-4 are labeled 1; whereas digits 5-9 are labeled 0. Additionally, 25% label noise is added, i.e., the labels are flipped with probability 0.25, both at train and test time, capping the achievable test accuracy to 75%. In this dataset, each digit is RGB colored. During training, label 1 is given the color green with probability 0.9 and red with probability 0.1. On the other hand, label 0 is given red color with probability 0.9 and green with probability 0.1. This introduces a high degree of spurious correlation between color and the label. Thus, ERM is expected to significantly overfit to color for predicting the label.

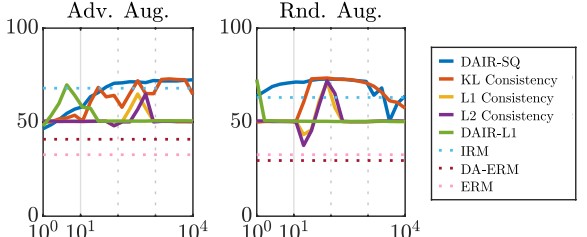

Figure 6: Test accuracy vs $\lambda$ on Colored MNIST for Adversarial Color aug. and Random Color aug.

At test time, the correlation with color is reversed for digits. Hence, vanilla ERM is expected to perform worse than 50% coin flip at test time. We explore two data augmentation schemes in this experiment. For the *Adversarial Augmentation (Adv. Aug.)* setup, the augmented images will have their color flipped (from red to green or vice versa) with probability 0.1. For the *Random Augmentation (Rnd. Aug.)* setup, the augmented images are colored uniformly at random. Detailed description of the setup and additional experiments can be found in Appendix D and Appendix E.1, respectively.

Figure 6 suggests that DAIR-SQ and KL consistency regularization achieve ∼72% test accuracy using both augmentation schemes, outperforming the state-of-the-art 68% test accuracy reported by IRM (Arjovsky et al., 2019), and almost reaching the 75% cap. We also compare to more recent baselines in Table 15 (Appendix E.1). We note however that this comparison may be unfair because IRM does not have access to any pairing information between the original and the augmented samples. As we observe in the next section, such information is readily available in several real-world benchmarks and DAIR-SQ can exploit it to achieve new state-of-the-art results. We also notice that neither variant of DA-ERM achieves test performance better than 50% coin flip in this experiment, while Adversarial Augmentation seems to fare better than Random Augmentation.

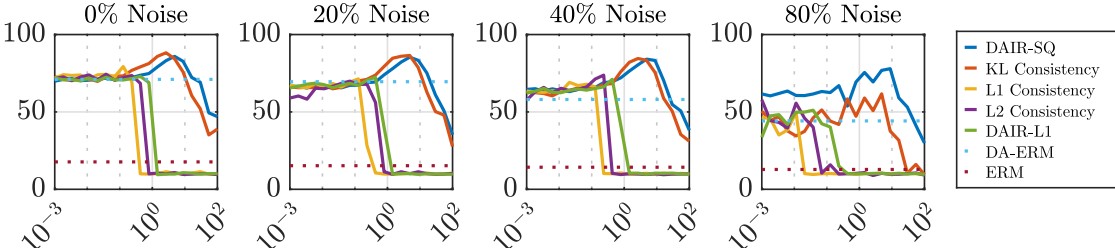

Figure 7: Test accuracy as a function of $\lambda$ for different label noise levels for **Weak Rotation** augmentation.

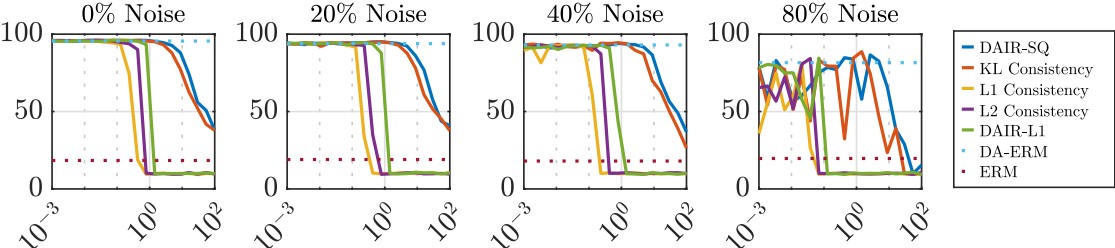

Figure 8: Test accuracy as a function of $\lambda$ for different label noise levels for **Strong Rotation** augmentation.

## 4.2 Rotated MNIST

Rotated MNIST (Ghifary et al., 2015) is a dataset where MNIST digits are rotated. We work with two different sets of degrees of rotation for Rotated MNIST. The first one is *Weak Rotation* where the digits are rotated uniformly at random $[0, \frac{\pi}{6})$ radians. In *Strong Rotation* the digits are rotated uniformly at random $[0, 2\pi)$ radians. To evaluate the robustness of the methods, we further add label noise at training time where the label is replaced with a digit chosen from $\{0,\dots,9\}$ uniformly at random with a certain probability. No label noise is added at test time. Detailed setup is in Appendix D. Model architecture and training parameters are detailed Appendix D.

In the first experiment, we use Weak Rotation for data augmentation while at test time we use Strong Rotation. Thus, some test time rotations have not been observed at training time. Figure 7 shows the test performance of all algorithms (averaged over three runs) as a function of $\lambda$. As can be seen, ERM (with no data augmentation) does not generalize to rotated test images and performs poorly. DA-ERM offers significant performance improvement over ERM. When $\lambda$ is very small all variants of consistency regularization are virtually the same as DA-ERM. DAIR-SQ and KL regularizer outperform other regularizers and are the only two variants that offer improvement over DA-ERM as $\lambda$ increases. As the label noise level becomes larger, DAIR-SQ is more robust than KL regularizer and offers the best performance.

Besides DAIR-SQ and KL regularizer, it is noteworthy that the other consistency regularization variants did not offer improvement over DA-ERM and they converged to poor local minima with 10% test accuracy (random) for large $\lambda$. We were not able to remedy this by tuning of their step size. See Section 2.2.1 for further justification of this phenomenon. We also observe that the performance of both DAIR-SQ and KL regularizer achieves a sweet spot for some finite $\lambda$, i.e., the performance starts to drop for large values of $\lambda$. This is not theoretically expected and can be attributed to the practical issues with solving the consistency regularization problem. We further investigate this phenomenon in Appendix F and provide some explanations.

The setup for the second experiment is the same as the first one, except we also use Strong Rotation in training for augmentation, so there is no test-time distribution shift for DA-ERM. As can be seen in Figure 8, data augmentation achieves very good performance in this case and none of the DAIR regularizers offer any improvement beyond data augmentation. We suspect this to be true in general; if the data augmentation is well-devised and optimized the resulting model could become invariant to the desired transformations at test time. Additional experiments on consistency metric can be found in Appendix E.1.

### 4.3 Invariant Visual Question Answering

In this experiment, we focus on the IV-VQA dataset which contains one or more edited images constructed from orignal images by removing an object which is irrelevant to answering the question. A robust model should make consistently predict the same answer for the original and corresponding edited images.

Similar to the setup in CV-VQA, We use the SAAA (Kazemi & Elqursh, 2017) model and compare performance against baselines using accuracy and consistency metrics.

| Algorithm | VQA v2 val ↑ | CM ↑ | Predictions Flipped ↓ | pos → neg ↓ | neg → pos ↓ | neg → neg ↓ |
|---|---|---|---|---|---|---|
| ERM (Kazemi & Elqursh, 2017) | 64.18 | 0.9456 | 8.64 | 3.45 | 3.00 | 2.2 |
| DA-ERM (Agarwal et al., 2020) | 64.66 | 0.9543 | 7.47 | 2.92 | 2.73 | 2.73 |
| KL Feature Consistency | 64.57 | 0.9582 | 7.07 | 2.73 | 2.50 | 1.84 |
| DAIR | **64.75** | **0.9606** | **6.33** | **2.54** | **2.22** | **1.57** |

Table 4: Accuracy and Consistency metrics on VQA v2 val & IV-VQA test set.

The results are reported in Table 4. In addition to our CM score, we borrow the consistency metrics from (Agarwal et al., 2020) that measure three types of flips namely, pos → neg, neg → pos and neg → neg. A pos → neg flip indicates that the answer predicted with the original image was correct but was wrong with the corresponding edited image. A neg → neg flip indicates that the answer changes from original to edited image but is wrong for both. DAIR achieves a higher accuracy as compared to all baselines across both datasets, while improving under CM score and the 'Predictions flipped' metric which is the sum of the three types of flips. While applying DAIR to this task, we observe a trade-off between the VQA accuracy and the consistency metrics controlled by the $\lambda$ parameter. By increasing $\lambda$, the consistency score rises to as high as 98%, albeit sacrificing the VQA accuracy by 6-7%. For moderate values of $\lambda$, DAIR maintains classification accuracy of the model while enforcing consistency across variations in the visual space. See Appendix I for more details.

### 4.4 Training robust deep networks against adversarial attacks

Neural networks have been widely used in various applications, especially in computer vision. However, neural networks are vulnerable to adversarial attacks, such as Fast Gradient Sign Method (FGSM) (Goodfellow et al., 2014) and Projected Gradient Descent (PGD) (Madry et al., 2018), where small adversarial perturbations in the input are designed to significantly alter the output prediction.

In this section, we consider training robust neural networks against adversarial attacks and compare with state-of-the-art baseline models which are specifically designed for this task. We evaluate the performance of each algorithm against PGD attacks as well as the clean (no attack-free) accuracy. In our approach, the augmented examples

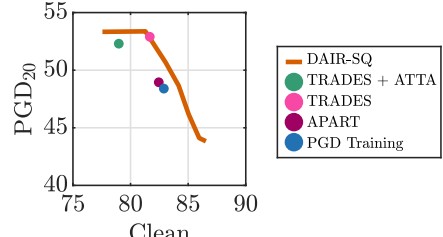

Figure 9: PGD20/Clean Acc. trade-off by sweeping $\lambda$.

$\tilde{z}$ can be generated by a certain strong attack, such as Projected Gradient Descent (PGD) or CW (Carlini & Wagner, 2017).We conduct our experiments on CIFAR-10 dataset and compare our approach with several other state-of-the-art baselines.

The performance of our algorithm against the Fast Gradient Sign Method (FGSM) and variants of PGD, is summarized in Table 5, which shows that our results are competitive with the baselines. We report the performance of DAIR in Table 5 based on the configurations that give the best Clean accuracy followed by the best Robust accuracy against PGD20 in parenthesis afterward. The trade-off curve shown in Figure 9 suggests that by sweeping the value of $\lambda$, DAIR can achieve a better clean accuracy but a slightly lower PGD20 accuracy, and dominates most of the baseline, while it achieves a similar performance with TRADES. Note that the formulation in TRADES is equivalent to consistency regularization with KL divergence between the logits of the original and adversarial images. As opposed to our setup, the regularizer term in TRADES is also used in solving the maximization problem to generate adversarial images, whereas we only use the original loss for generating the adversarial examples.

| # | Algorithm | Clean (%) | FGSM (%) | PGD20 (%) | CM (%) |
|---|-----------|-----------|----------|-----------|--------|
| 1 | PGD Training (Madry et al., 2018) | 82.89 | 55.38 | 48.40 | – |
| 2 | APART (Li et al., 2020) | 82.45 | 55.33 | 48.95 | 60.05 |
| 3 | DAIR ($\lambda = 6$) | 83.04 | 57.57 | 50.68 | 62.66 |
| 4 | TRADES + ATTA (Zheng et al., 2020) | 78.98 | 55.58 | 52.30 | 60.56 |
| 5 | TRADES (Zhang et al., 2019) | 81.67 | 57.78 | 52.90 | 63.14 |
| 6 | DAIR ($\lambda = 16.7$) | 81.29 | 58.58 | 53.37 | 67.51 |

Table 5: CIFAR-10 test accuracies under no attack (clean), FGSM, and PGD20 attacks, and accuracy consistency metric between original and PGD20 attack.

We also report the accuracy consistency metric (CM) in this experiment in Table 5. CM captures the consistency of accuracy on PGD20 attack compared to clean examples. We observe that DAIR outperforms all baselines, which is in line with its best generalization to different attacks.

Lastly, we compare DAIR with more recent baselines (namely Tack et al. (2021)) on a different robust classification against adversarial attacks setup, where we observe that DAIR offers competitive performance improvements compared to the state-of-the-art baselines (see Appendix K).

### 4.5   Robust regression: simultaneous domain shift and label noise

In this experiment, we consider a regression task to minimize the root mean square error (RMSE) of the predicted values on samples from the Drug Discovery dataset. The task is to predict the bioactivities given a set of chemical compounds (binary features). We follow the setup of Li et al. (2021) to introduce random noise to corrupt the targets. Furthermore, similar to Colored MNIST, we add a spurious binary feature to the original setup. At training time, the spurious feature is set to 1 if the target is above a threshold (the median of all the targets in the training samples), and 0 otherwise. At test time, this condition is reversed leading to poor generalization. We compare using ERM, DA-ERM and DAIR formulations under 0%, 20% and 40% noise levels on three baselines: $\mathcal{L}_2$ loss, Huber loss, and negatively tilted loss (Li et al., 2021), which is called tilted empirical risk minimization (TERM) and is designed for robust regression. For each of these baselines, we perform data augmentation by randomly assigning the spurious feature as 0 or 1 with equal probability. Finally, we apply the DAIR regularizer to each of these loss functions with $\lambda = 10$.

| Algorithms | Test RMSE (Drug Discovery dataset) | | | | | | | | | |
|---|---|---|---|---|---|---|---|---|---|---|
| | 0% Noise | | | 20% Noise | | | 40% Noise | | | Clean |
| | ERM | DA-ERM | DAIR | ERM | DA-ERM | DAIR | ERM | DA-ERM | DAIR | ERM |
| $\mathcal{L}_2$ loss | 1.97 (0.00) | 1.36 (0.00) | **1.23** (0.00) | 4.33 (0.04) | 2.52 (0.05) | 2.04 (0.06) | 5.30 (0.04) | 3.47 (0.07) | 2.99 (0.09) | 1.23 (0.00) |
| Huber (Huber, 1964) | 1.84 (0.00) | 1.27 (0.00) | 1.24 (0.00) | 2.93 (0.05) | 1.50 (0.02) | 1.39 (0.02) | 4.40 (0.07) | 2.18 (0.04) | 1.70 (0.05) | **1.16** (0.00) |
| TERM (Li et al., 2021) | 1.74 (0.00) | 1.26 (0.00) | 1.25 (0.00) | 1.87 (0.01) | **1.27** (0.01) | **1.27** (0.01) | 2.01 (0.02) | **1.33** (0.01) | **1.31** (0.01) | 1.23 (0.00) |

Table 6: Test RMSE for varying degrees of label noise for ERM, DA-ERM, and DAIR using different losses.

The results of this experiment are reported in Table 6. In the last column of the table we report results on the clean dataset without any spurious features for comparison purposes. As can be seen, without data augmentation all methods fall prey to spurious features and perform poorly, especially as the noise level is increased. It is noteworthy that while TERM is not designed for domain shift, it slightly outperforms the other baselines in the presence of spurious features showing that TERM has some inherent robustness to the domain shift. By adopting data augmentation, testing error decreases but is still quite large as compared to the Clean ERM setup for high values of noise. Notably, DAIR is able to reduce the testing error across all objectives and noise levels with the gap between DAIR and other approaches increasing with the degree of noise. For the noiseless setup, DAIR is able to almost recover the Clean ERM accuracy for all three objectives. The gains achieved with DAIR are prominent for $\mathcal{L}_2$ and Huber, but marginal for TERM. Finally, DAIR combined with TERM can simultaneously handle domain shift and noisy labels as can be seen in this table.

### 4.6 ImageNet-9 background challenge

Deep learning models have outperformed human-level performance in many applications of which the most prominent is image classification. However, these models are extremely vulnerable to overfitting to spurious correlations such as background features. ImageNet-9 Background Challenge (Xiao et al., 2020) was proposed to test the background robustness of image classification models. In this challenge, seven variations of images such as background/foreground removal, are provided to measure the extent to which models rely on the background. For example, variations Only-BG-B and Only-BG-T remove the backgrounds and therefore the test accuracy is expected to be low for a model which does not learn spurious background features. See (Xiao et al., 2020, Figure 1) for example images of each variation. Xiao et al. (2020) choose to train a model on Mixed-Rand variation, which is the most powerful and comprehensive augmentation scheme, and demonstrated that the resulting model is more robust. We choose the Mixed-Rand variation as our augmentation scheme and compare the test accuracy on all seven variations of the models trained by ERM, (Xiao et al., 2020), DA-ERM, DAIR and KL.

Table 7 summarizes the results. We see that DAIR outperforms ERM and DA-ERM by a large margin and is similarly competitive as KL feature consistency regularization (see the third column). In particular, DAIR outperforms DA-ERM on all metrics for which a higher test accuracy is more desirable. DAIR improves performance on the variations which include domain shift, such as Mixed-Same, Mixed-Next and Only-FG. In particular, it also helps Original and the Mixed-Rand variations which are seen during training as well. In this experiment, similar to Section 3.1, DAIR not only enhances out-of-domain generalizability/robustness but also gives the best the in-distribution performance (original). The detailed training setup can be found in Appendix L

| | Original ↑ | Mixed-Rand ↑ | Mixed-Same ↑ | Mixed-Next ↑ | Only-BG-B ↓ | Only-BG-T ↓ | No-FG | Only-FG ↑ |
|---|---|---|---|---|---|---|---|---|
| ERM (Original) | 69.43 | 38.57 | 59.63 | 34.27 | 25.83 | 31.06 | 37.04 | 44.00 |
| (Xiao et al., 2020) | 49.41 | 51.98 | 51.85 | 50.48 | 13.41 | **13.80** | 20.05 | 52.22 |
| DA-ERM | 64.02 | 58.05 | 58.81 | 48.17 | 16.91 | 22.27 | 28.44 | 56.62 |
| KL Feature Consistency | 70.84 | **65.36** | 68.79 | **64.07** | **13.23** | 20.81 | 26.77 | **67.48** |
| DAIR | **72.91** | 63.48 | **69.16** | 62.02 | 17.19 | 24.02 | 31.88 | 66.15 |

Table 7: ImageNet-9 Backgrounds Challenge test accuracy on different shifted test sets. We use Mixed-Rand as augmentation during training for DA-ERM, DAIR, and KL feature consistency. Arrows next to the heading of each column indicate the desired direction of the metric. For example, the accuracy on Original images should be as high as possible but the accuracy on Only-BG-B should be as low as possible.

## 5 Conclusion

In this paper, we proposed a simple yet effective consistency regularization technique, called data augmented loss invariant regularization (DAIR). DAIR is applicable when data augmentation is used to promote performance invariance across pairs of original and augmented samples, and it enforces the loss to be similar on the original and the augmented samples. We provided motivation and justification for DAIR, and particularly showed that it can recover the optimal solution in a certain regression task where data augmentation alone is insufficient. We also compared DAIR with other consistency regularizers and showed that existing consistency regularization techniques cannot handle covariant data augmentation. On the other hand, DAIR is broadly applicable to tasks involving invariant/covariant data augmentation. We empirically evaluated DAIR in five real-world machine learning tasks, namely task-oriented dialog modeling, visual question answering, robust regression, robust classification against adversarial attacks, and ImageNet-9 background challenge. Our experiments confirmed a major benefit of DAIR as some other consistency regularizers cannot be applied broadly. Empirically, DAIR set new state-of-the-art results in dialog state tracking and VQA benchmarks which involved covariant data augmentation, and provided competitive results in all other benchmarks. This demonstrated the effectiveness of DAIR despite its simplicity.

Several problems remain open for future research: An in-depth theoretical understanding of the properties of DAIR that lead to its superior empirical performance on broad applications is an important open question.

Further, automated hyperparameter tuning techniques for the strength of the regularizer is another avenue for future research. Finally, a discussion on limitations and broader impact appears in Section 6.

## 6 Limitations & Broader Impact

Firstly, while we demonstrated the success of DAIR when the pairing information between original and augmented training samples is known, the applicability of DAIR remains limited in only setups where such pairing information is available. We also remark that DAIR incurs double the computational cost compared with algorithms which only consider one sample for backpropagation (e.g., only an augmented examples) rather than the pair.

Secondly, applying DAIR to arbitrary tasks requires domain knowledge about the nature of the desired invariances to be promoted which is expressed by choosing appropriate augmented samples. In particular, we did not address devising good data augmentation procedures, but rather we argued that if data augmentation is already employed (i.e. DA-ERM is used), DAIR can lead to remarkable gains (almost) with marginally added cost via further regularization of the losses. It remains to examine whether DAIR could be used as a component for solving domain generalization for arbitrary domain shifts where data augmentation pairing cannot be performed in a straightforward fashion.

Thirdly, we demonstrated the effectiveness of DAIR on a variety of supervised tasks involving multimodal, generative and regression models. However, the applicability of DAIR to semi-supervised or self-supervised learning remains to be seen.

Finally, while we showed that DAIR boosts existing performance metrics, such as accuracy, the interplay of DAIR with other important socially consequential metrics, such as group fairness and privacy, was not explored in this paper. It remains to be seen whether DAIR may have any positive or negative consequences on these other Responsible AI metrics.

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

## Appendix

In this appendix, we provide additional linear regression example with theoretical analysis (Appendix A) and its extension (Appendix B); include proofs of the analyses from Section 2 (Appendix C); details on MNIST experiments (Appendices D to E); practical considerations when DAIR used in training (Appendix F); the impact of partial augmentation (Appendix G); training details on experiments in Sections 3 and 4 (Appendices H to L). We provide a table of contents below for easier navigation.

## Contents

# A   Additional linear regression example with theoretical analysis

In this section, we answer the question "What does DAIR offer beyond DA-ERM?" both theoretically and empirically through a simple toy example. In this example, we demonstrate that DAIR can fundamentally outperform DA-ERM, even in the limit of infinite training samples (no overfitting due to finite samples). Consider a linear regression problem where at the training time the input is $\mathbf{x}_{\text{train}} = (x, s = y)$ and the label $y$, i.e., $z_{\text{train}} = (\mathbf{x}_{\text{train}}, y)$. Here, $x \sim \mathcal{N}(0, \sigma_x^2)$, and $y = x + \varepsilon$, where $\varepsilon$ is independent of $x$ and $\varepsilon \sim \mathcal{N}(0, \sigma_\varepsilon^2)$. In this example, the target is explicitly provided as a spurious feature to the learner at the training time. At test time, the spurious feature is absent, i.e., $\mathbf{x}_{\text{test}} = (x, s = 0)$.

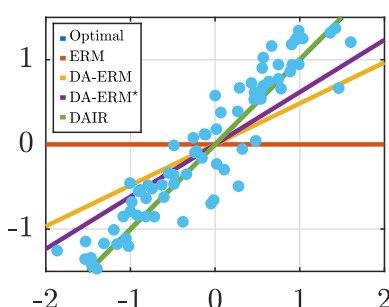

Figure 10: The plot of the optimal, ERM, DA-ERM and DAIR-SQ ($\lambda = 100$) regressors for the toy example of Appendix A.

Clearly, in this toy example, the optimal regressor is $w^\star = (w_1^\star, w_2^\star)^\top = (1, 0)^\top$. However, absent the knowledge of the spurious feature vanilla ERM will learn $w_{\text{ERM}} \approx (0, 1)^\top$, completely overfitting the spurious feature. We assume that the learner has access to a data augmentation module that generates $\widetilde{z} = A(z; a, \sigma_n^2) = (\mathbf{x}_{\text{aug}}, y)$, such that $\mathbf{x}_{\text{aug}} = (x, s = ay + n)$ where $n \sim \mathcal{N}(0, \sigma_n^2)$. The augmented data will encourage the learned model to become invariant to the spurious feature. In Figure 10, we perform simulations with $a = 0.5$, $\sigma_x^2 = 1$, $\sigma_\varepsilon^2 = 0.25$, $\sigma_n^2 = 0.1$ and plot four linear regressors associated with the slope of their respective $w_1$. We ignore $w_2$ as the second spurious feature is absent at test time and hence $w_2$ does not impact test performance. The optimal regressor is shown as the blue line, with a slope of 1. ERM (red line) completely fails due to the overfitting to the spurious feature. DA-ERM (orange line) significantly improves over ERM but still is far from optimal performance. DA-ERM$^\star$ (purple line) which is solely trained on augmented examples (ignoring the original examples) slightly outperforms DA-ERM but still significantly overfits to the spurious feature. DAIR-SQ (green line) almost recovers the optimal solution. This is not a coincidence. We prove that DAIR-SQ is optimal for a class of linear regression problems, while DA-ERM does not approach optimal performance even in the limit of infinite samples. Here we state the rigorous statement, followed by proof.

**Proposition 5.** *Consider a linear regression problem with training point $z_{train} = (\mathbf{x}_{train}, y)$ where $\mathbf{x}_{train} = (x, s = y)$; $y$ denotes label and $s$ denotes the spurious feature. Here, $x \sim \mathcal{N}(0, \sigma_x^2)$, and $y = x + \varepsilon$, where $\varepsilon$ is independent of $x$ and $\varepsilon \sim \mathcal{N}(0, \sigma_\varepsilon^2)$. Assume the learner has access to a data augmentation module which generates $z_{aug} = (\mathbf{x}_{aug}, y)$ where $\mathbf{x}_{aug} = (x, s = ay + n)$. Here $n \sim \mathcal{N}(0, \sigma_n^2)$, and $a \in \mathbb{R}$. At test time, the spurious feature is absent, i.e., $\mathbf{x}_{test} = (x, s = 0)$. Both DAIR-SQ and DA-ERM are applied to solve this problem: DAIR-SQ achieves optimal test error as number of samples grows and $\lambda \to \infty$. On the other hand, DA-ERM cannot generally recover optimal performance even in the limit of infinite training data.*

*Proof.* First let us present the DA-ERM solution:

$$
\begin{aligned}
f_{\text{DA-ERM}}(w) =& \mathbb{E}\left[(w_1 x + w_2 y - y)^2 + (w_1 x + w_2(ay + n) - y)^2\right] && (1) \\
=& \mathbb{E}\left[w_1^2 x^2 + (w_2 - 1)^2 y^2 + 2w_1(w_2 - 1)xy\right] \\
& + \mathbb{E}\left[w_1^2 x^2 + (w_2 a - 1)^2 y^2 + w_2^2 n^2\right] \\
& + \mathbb{E}\left[2w_1(w_2 a - 1)xy + 2w_1 w_2 xn + 2w_2(w_2 a - 1)yn\right] && (2) \\
=& w_1^2 \sigma_x^2 + (w_2 - 1)^2(\sigma_x^2 + \sigma_\varepsilon^2) + 2w_1(w_2 - 1)\sigma_x^2 \\
& + w_1^2 \sigma_x^2 + (w_2 a - 1)^2(\sigma_x^2 + \sigma_\varepsilon^2) + w_2^2 \sigma_n^2 \\
& + 2w_1(w_2 a - 1)\sigma_x^2 && (3) \\
=& (w_1 + w_2 - 1)^2 \sigma_x^2 + (w_2 - 1)^2 \sigma_\varepsilon^2 \\
& + (w_1 + w_2 a - 1)^2 \sigma_x^2 + (w_2 a - 1)^2 \sigma_\varepsilon^2 + w_2^2 \sigma_n^2. && (4)
\end{aligned}
$$

Hence, the solution of $w_{\text{DA-ERM}}^\star = \arg\min_w f_{\text{DA-ERM}}(w)$ is given by

$$
2w_1^\star + (1 + a)w_2^\star - 2 = 0,
$$
$$
(w_1^\star + w_2^\star - 1)\sigma_x^2 + (w_2^\star - 1)\sigma_\varepsilon^2 + a(w_1^\star + w_2^\star a - 1)\sigma_x^2 + a(w_2^\star a - 1)\sigma_\varepsilon^2 + w_2^\star \sigma_n^2 = 0. \quad (5)
$$

Subsequently,

$$w^\star_{\text{DA-ERM}} = \begin{pmatrix} \frac{a^2(\sigma_x^2+\sigma_\varepsilon^2)-2a(\sigma_x^2+\sigma_\varepsilon^2)+\sigma_x^2+\sigma_\varepsilon^2+2\sigma_n^2}{a^2(\sigma_x^2+2\sigma_\varepsilon^2)-2a\sigma_x^2+\sigma_x+2(\sigma_\varepsilon^2+\sigma_n^2)} \\ \\ \frac{2(a+1)\sigma_\varepsilon^2}{a^2(\sigma_x^2+2\sigma_\varepsilon^2)-2a\sigma_x^2+\sigma_x^2+2(\sigma_\varepsilon^2+\sigma_n^2)} \end{pmatrix}. \tag{6}$$

$$\begin{aligned} w^\star_{\text{DAIR}} &= \arg\min_w f_{\text{DAIR}}(w) \\ &= \arg\min_w \mathbb{E}\left[(w_1 x + w_2 y - y)^2 + (w_1 x + w_2(ay+n) - y)^2\right] \\ &\quad + \left[\lambda(|w_1 x + w_2 y - y| - |w_1 x + w_2(ay+n) - y|)^2\right]. \end{aligned}$$

When $\lambda \to \infty$, we have $w^\star_{\text{DAIR},2} = 0$ and hence:

$$w^\star_{\text{DAIR}} = \begin{pmatrix} 1 \\ 0 \end{pmatrix}.$$

We then evaluate the testing loss assuming the spurious feature is absent, i.e., $\mathbf{x}_{\text{test}} = (x, s = 0)$.

$$\begin{aligned} \ell_{\text{DAIR}}(\mathbf{x}_{\text{test}}; w^\star_{\text{DAIR}}) &= \mathbb{E}\left[(w^{\star\top}_{\text{DAIR}}\mathbf{x}_{\text{test}} - y)^2\right] \\ &= \mathbb{E}\left[(x - (x+\varepsilon))^2\right] \\ &= \sigma_\varepsilon^2. \end{aligned}$$

$$\begin{aligned} \ell_{\text{DA-ERM}}(\mathbf{x}_{\text{test}}; w^\star_{\text{DA-ERM}}) &= \mathbb{E}\left[(w^{\star\top}_{\text{DA-ERM}}\mathbf{x}_{\text{test}} - y)^2\right] \\ &= \mathbb{E}\left[\left(\frac{a^2(\sigma_x^2+\sigma_\varepsilon^2)-2a(\sigma_x^2+\sigma_\varepsilon^2)+\sigma_x^2+\sigma_\varepsilon^2+2\sigma_n^2}{a^2(\sigma_x^2+2\sigma_\varepsilon^2)-2a\sigma_x^2+\sigma_x+2(\sigma_\varepsilon^2+\sigma_n^2)}x - (x+\varepsilon)\right)^2\right] \\ &= \sigma_\varepsilon^2 + \frac{(a+1)^4\sigma_\varepsilon^4\sigma_x^2}{(a^2(\sigma_x^2+2\sigma_\varepsilon^2)-2a\sigma_x^2+\sigma_x+2(\sigma_\varepsilon^2+\sigma_n^2))^2} \\ &\geq \ell_{\text{DAIR}}, \end{aligned}$$

completing the proof. $\qquad\qquad\square$

One can show that simple data independent regularization methods (e.g. weight decay) cannot help close the gap between the performance of DA-ERM and DAIR (see Proposition 6). While the toy example presented an extreme case with a spurious feature equal to the output, we prove theoretically that the same conclusion holds as long as a subset of features have different correlation patterns with the output at training and test time (see Proposition 7 for the general multi-variate linear regression setup). Note that in this toy example when $\sigma_n \to \infty$, DA-ERM could also recover $w^\star$. One can interpret that as $\sigma_n \to \infty$, the augmenter becomes stronger and forces $w_2$ to vanish. On the other hand, DAIR recovers $w^\star$ with a much weaker augmenter. This is crucial since in real-world applications, designing strong augmentation schemes requires careful design. See the example of *Weak Rotation* setup in Section 4.2 for details.

**Proposition 6.** *Consider the case in which a weight decay regularizer $\frac{\gamma}{2}(w_1^2 + w_2^2)$ is added to the DA-ERM, the resulting solution is the following:*

$$w^\star_{DA\text{-}ERM\text{-}WD} = \begin{pmatrix} \frac{a^2(\sigma_\varepsilon^2+\sigma_x^2)-2a(\sigma_\varepsilon^2+\sigma_x^2)+2\gamma+\sigma_\varepsilon^2+2\sigma_n^2+\sigma_x^2}{a^2(\gamma(\sigma_\varepsilon^2+\sigma_x^2)+2\sigma_\varepsilon^2+\sigma_x^2)-2a\sigma_x^2+\gamma^2+\gamma(\sigma_\varepsilon^2+\sigma_n^2+\sigma_x^2+2)+2\sigma_\varepsilon^2+2\sigma_n^2+\sigma_x^2} \\ \\ \frac{(a+1)(\gamma(\sigma_\varepsilon^2+\sigma_x^2)+2\sigma_\varepsilon^2)}{a^2(\gamma(\sigma_\varepsilon^2+\sigma_x^2)+2\sigma_\varepsilon^2+\sigma_x^2)-2a\sigma_x^2+\gamma^2+\gamma(\sigma_\varepsilon^2+\sigma_n^2+\sigma_x^2+2)+2\sigma_\varepsilon^2+2\sigma_n^2+\sigma_x^2}) \end{pmatrix}.$$

*Proof of Proposition 6.* The proof follows the same idea of Proposition 5 and therefore it is omitted here. $\qquad\square$

Proposition 6 shows that even using the weight decay regularizer would not close the gap between the performance of DA-ERM and DAIR. In other words $w^\star_{\text{DA-ERM-WD}} \neq w^\star = (1,0)$ unless $\sigma_n^2 \to \infty$ and $\gamma = 0$.

# B  Multi-dimensional extension of linear regression example (Appendix A)

Consider a multi-dimensional extension of the example in Appendix A. During training, the input is $\mathbf{x}_{\text{train}} = [x\ s_{\text{train}}]^\top \in \mathbb{R}^{d+k}$ and $y = \mathbf{1}^\top x + \varepsilon$, where $x \sim \mathcal{N}(0, \sigma_x^2 I) \in \mathbb{R}^d$, $\varepsilon \sim \mathcal{N}(0, \sigma_\varepsilon^2)$, $s_{\text{train}} = yv_{\text{train}} + n_{\text{train}}$, $v_{\text{train}} \in \mathbb{R}^k$ and $n_{\text{train}} \sim \mathcal{N}(0, \sigma_{n_{\text{train}}}^2 I) \in \mathbb{R}^k$. Similar to the toy example, we introduce spurious feature $s_{\text{train}}$ to mislead the model. ERM based approach should overfit to use the information in $s_{\text{train}}$ if $\sigma_{n_{\text{train}}}^2$ is small. Suppose the learner also has access to the augmented datapoints of the form $\mathbf{x}_{\text{aug}} = [x\ s_{\text{aug}}]^\top \in \mathbb{R}^{d+k}$, where $s_{\text{aug}} = yu_{\text{aug}} + n_{\text{aug}}, u_{\text{aug}} \in \mathbb{R}^k$, and $n_{\text{aug}} \sim \mathcal{N}(0, \sigma_{n_{\text{aug}}}^2 I) \in \mathbb{R}^k$. The augmented data points will encourage the model to be invariant to the spurious feature during training. The testing data is $\mathbf{x}_{\text{test}} = [x\ \mathbf{0}]^\top \in \mathbb{R}^{d+k}$. Again, the optimal $w$ is $[\mathbf{1}\ \mathbf{0}]^\top$ and we compare the performances of DA-ERM and DAIR. One can think of this is a simplified setup that is aimed at emulating a case with some features, e.g., background, bearing no impact on the labels, whereas they may be (highly) correlated with the label during the training. One may guess the results of comparison as this is the extension of the toy example in the main body of the paper: DAIR mostly works better than DA-ERM. We introduce the formal proposition below.

**Proposition 7.** *Consider the linear least squares regression problem for predicting the target variable $y$ in the problem described above. Assume that the learner has access to a data augmentation module that perturbs the spurious feature, as described above. Consider the population level loss (i.e. number of samples is infinity). Then, for any value of $\sigma_{n_{train}}^2$, $\sigma_{n_{aug}}^2$, $v_{train}$ and $n_{aug}$, DAIR-SQ achieves optimal test error as $\lambda \to \infty$. On the other hand, DA-ERM cannot obtain optimal performance (other than for only certain corner cases such as $\sigma_{n_{aug}}^2 \to \infty$ and/or $\sigma_{n_{train}}^2 \to \infty$).*

*Proof.* Assuming the linear least squares regression fit, the objective function of DA-ERM can be written as

$$
\begin{aligned}
2\mathbb{E}[f_{\text{DA-ERM}}(w)] &= \mathbb{E}\left[(w^\top \mathbf{x}_{\text{train}} - y)^2 + (w^\top \mathbf{x}_{\text{aug}} - y)^2\right] \\
&= \mathbb{E}\left[(w_1^\top x + w_2^\top s_{\text{train}} - y)^2 + (w_1^\top x + w_2^\top s_{\text{aug}} - y)^2\right] \\
&= \mathbb{E}\left[(w_1^\top x + w_2^\top (v_{\text{train}} y + n_{\text{train}}) - y)^2 + (w_1^\top x + w_2^\top (u_{\text{aug}} y + n_{\text{aug}}) - y)^2\right] \\
&= \mathbb{E}\left[(w_1^\top x + w_2^\top (v_{\text{train}}(\mathbf{1}^\top x + \varepsilon) + n_{\text{train}}) - \mathbf{1}^\top x - \varepsilon)^2\right] \\
&\quad + \mathbb{E}\left[(w_1^\top x + w_2^\top (u_{\text{aug}}(\mathbf{1}^\top x + \varepsilon) + n_{\text{aug}}) - \mathbf{1}^\top x - \varepsilon)^2\right] \\
&= \mathbb{E}\left[(x^\top(w_1 + \mathbf{1}v_{\text{train}}^\top w_2 - \mathbf{1}) + \varepsilon(w_2^\top v_{\text{train}} - 1) + n_{\text{train}}^\top w_2)^2\right] \\
&\quad + \mathbb{E}\left[(x^\top(w_1 + \mathbf{1}u_{\text{aug}}^\top w_2 - \mathbf{1}) + \varepsilon(w_2^\top u_{\text{aug}} - 1) + n_{\text{aug}}^\top w_2)^2\right] \\
&= \sigma_x^2 \|w_1 + \mathbf{1}v_{\text{train}}^\top w_2 - \mathbf{1}\|^2 + \sigma_\varepsilon^2 (w_2^\top v_{\text{train}} - 1)^2 + \sigma_{n_{\text{train}}}^2 \|w_2\|^2 \\
&\quad + \sigma_x^2 \|w_1 + \mathbf{1}u_{\text{aug}}^\top w_2 - \mathbf{1}\|^2 + \sigma_\varepsilon^2 (w_2^\top u_{\text{aug}} - 1)^2 + \sigma_{n_{\text{aug}}}^2 \|w_2\|^2,
\end{aligned}
$$

where $w = [w_1^\top w_2^\top]^\top$ with $w_1 \in \mathbb{R}^d$ and $w_2 \in \mathbb{R}^k$. Expanding the norms will result in

$$
\begin{aligned}
2\mathbb{E}[f_{\text{DA-ERM}}(w)] &= \sigma_x^2[w^\top \widehat{I} w + 2w^\top \widehat{\mathbf{1}}\tilde{v}_{\text{train}}^\top w - 2\widehat{\mathbf{1}}^\top w + w^\top \tilde{v}_{\text{train}}\widetilde{\mathbf{1}}^\top \widetilde{\mathbf{1}}\tilde{v}_{\text{train}}^\top w - 2\widetilde{\mathbf{1}}^\top \widetilde{\mathbf{1}}\tilde{v}_{\text{train}}^\top w + \widehat{\mathbf{1}}^\top \widehat{\mathbf{1}}] \\
&\quad + \sigma_\varepsilon^2[w^\top \tilde{v}_{\text{train}}\tilde{v}_{\text{train}}^\top w - 2w^\top \tilde{v}_{\text{train}} + 1] + \sigma_{n_{\text{train}}}^2 w^\top \widetilde{I} w \\
&\quad + \sigma_x^2[w^\top \widehat{I} w + 2w^\top \widehat{\mathbf{1}}\tilde{u}_{\text{aug}}^\top w - 2\widehat{\mathbf{1}}^\top w + w^\top \tilde{u}_{\text{aug}}\widetilde{\mathbf{1}}^\top \widetilde{\mathbf{1}}\tilde{u}_{\text{aug}}^\top w - 2\widetilde{\mathbf{1}}^\top \widetilde{\mathbf{1}}\tilde{u}_{\text{aug}}^\top w + \widehat{\mathbf{1}}^\top \widehat{\mathbf{1}}] \\
&\quad + \sigma_\varepsilon^2[w^\top \tilde{u}_{\text{aug}}\tilde{u}_{\text{aug}}^\top w - 2w^\top \tilde{u}_{\text{aug}} + 1] + \sigma_{n_{\text{aug}}}^2 w^\top \widetilde{I} w \\
&= w^\top[\sigma_x^2\widehat{I} + \sigma_x^2\tilde{v}_{\text{train}}\widehat{\mathbf{1}}^\top + \sigma_x^2\widehat{\mathbf{1}}\tilde{v}_{\text{train}}^\top + \sigma_x^2\tilde{v}_{\text{train}}\widetilde{\mathbf{1}}^\top \widetilde{\mathbf{1}}\tilde{v}_{\text{train}}^\top + \sigma_\varepsilon^2\tilde{v}_{\text{train}}\tilde{v}_{\text{train}}^\top + \sigma_{n_{\text{train}}}^2\widetilde{I}]w \\
&\quad + (-2\sigma_x^2\widehat{\mathbf{1}} - 2\sigma_x^2\tilde{v}_{\text{train}}\widetilde{\mathbf{1}}^\top \widetilde{\mathbf{1}} - 2\sigma_\varepsilon^2\tilde{v}_{\text{train}})^\top w \\
&\quad + w^\top[\sigma_x^2\widehat{I} + \sigma_x^2\tilde{u}_{\text{aug}}\widehat{\mathbf{1}}^\top + \sigma_x^2\widehat{\mathbf{1}}\tilde{u}_{\text{aug}}^\top + \sigma_x^2\tilde{u}_{\text{aug}}\widetilde{\mathbf{1}}^\top \widetilde{\mathbf{1}}\tilde{u}_{\text{aug}}^\top + \sigma_\varepsilon^2\tilde{u}_{\text{aug}}\tilde{u}_{\text{aug}}^\top + \sigma_{n_{\text{aug}}}^2\widetilde{I}]w \\
&\quad + (-2\sigma_x^2\widehat{\mathbf{1}} - 2\sigma_x^2\tilde{u}_{\text{aug}}\widetilde{\mathbf{1}}^\top \widetilde{\mathbf{1}} - 2\sigma_\varepsilon^2\tilde{u}_{\text{aug}})^\top w \\
&\quad + 2(\sigma_x^2\widehat{\mathbf{1}}^\top \widehat{\mathbf{1}} + \sigma_\varepsilon^2)
\end{aligned}
$$

where $\tilde{v}_{\text{train}} = [\mathbf{0}^\top \ v_{\text{train}}^\top]^\top$, $\tilde{u}_{\text{aug}} = [\mathbf{0}^\top \ u_{\text{aug}}^\top]^\top$, $\widetilde{I} = \begin{bmatrix} \mathbf{0} & \mathbf{0} \\ \mathbf{0} & I \end{bmatrix}^\top$, $\widehat{I} = \begin{bmatrix} I & \mathbf{0} \\ \mathbf{0} & \mathbf{0} \end{bmatrix}^\top$, $\widetilde{\mathbf{1}} = [\mathbf{0}^\top \ \mathbf{1}^\top]^\top$ and $\widehat{\mathbf{1}} = [\mathbf{1}^\top \ \mathbf{0}^\top]^\top$.

By optimality condition, we have:

$$
Qw_{\text{DA-ERM}}^\star = -b \tag{7}
$$

where

$$
\begin{aligned}
Q &= \sigma_x^2(\tilde{v}_{\text{train}}\widehat{\mathbf{1}}^\top + \widehat{\mathbf{1}}\tilde{v}_{\text{train}}^\top + \tilde{u}_{\text{aug}}\widehat{\mathbf{1}}^\top + \widehat{\mathbf{1}}\tilde{u}_{\text{aug}}^\top + \tilde{v}_{\text{train}}\widetilde{\mathbf{1}}^\top \widetilde{\mathbf{1}}\tilde{v}_{\text{train}}^\top + \tilde{u}_{\text{aug}}\widetilde{\mathbf{1}}^\top \widetilde{\mathbf{1}}\tilde{u}_{\text{aug}}^\top) \\
&\quad + 2\sigma_x^2\widehat{I} + (\sigma_{n_{\text{train}}}^2 + \sigma_{n_{\text{aug}}}^2)\widetilde{I} + \sigma_\varepsilon^2(\tilde{v}_{\text{train}}\tilde{v}_{\text{train}}^\top + \tilde{u}_{\text{aug}}\tilde{u}_{\text{aug}}^\top) \\
b &= \sigma_x^2(-2\widehat{\mathbf{1}} + (-\tilde{v}_{\text{train}} - \tilde{u}_{\text{aug}})\widetilde{\mathbf{1}}^\top \widetilde{\mathbf{1}}) + \sigma_\varepsilon^2(-\tilde{v}_{\text{train}} - \tilde{u}_{\text{aug}}).
\end{aligned}
$$

$\square$

One can check that $w_{\text{DA-ERM}}^\star \neq \begin{bmatrix} \mathbf{1} \\ \mathbf{0} \end{bmatrix}$ for generic choice of $v_{\text{train}}, u_{\text{aug}}, \sigma_x^2, \sigma_{n_{\text{aug}}}^2, \sigma_\varepsilon^2, \sigma_{n_{\text{train}}}^2$. This is because we have more free variables than the number of equations in equation 7. Thus, $w_{\text{DA-ERM}}^\star$ cannot recover the optimal regressor for the generic choice of parameters. More specifically, only in certain corner cases $w_{\text{DA-ERM}}^\star$ would recover the optimal regressor $\begin{bmatrix} \mathbf{1} \\ \mathbf{0} \end{bmatrix}$. For example, when $\sigma_{n_{\text{train}}}^2 \to +\infty$, we have $\frac{1}{\sigma_{n_{\text{train}}}^2}Q \to \widetilde{I}$ and $\frac{1}{\sigma_{n_{\text{train}}}^2}b \to \mathbf{0}$. Thus, in this corner case, the optimality condition equation 7 is asymptotically satisfied.

On the other hand, in the presence of DAIR regularizer, when $\lambda \to \infty$, the loss function in equation DAIR remains finite if and only if $\mathcal{R}(\ell(z; w), \ell(\tilde{z}; w)) = 0$, almost everywhere. Equivalently, the objective in equation DAIR remains finite (when $\lambda \to \infty$) if and only if

$$
(y - x^\top w_1 - s_{\text{train}}^\top w_2)^2 = (y - x^\top w_1 - s_{\text{aug}}^\top w_2)^2,
$$

for almost all realizations of data, which implies $w_2 = \mathbf{0}$. Thus, when $\lambda \to \infty$, $w_{\text{DAIR}}^\star \to [w_{1,\text{DAIR}}^{\star\top}, w_{2,\text{DAIR}}^{\star\top}]$ with $w_{2,\text{DAIR}}^\star = \mathbf{0}$. In other words, the coefficient of the spurious features vanishes as $\lambda \to \infty$ and hence the regression will recover the groundtruth regressor $\begin{bmatrix} \mathbf{1} \\ \mathbf{0} \end{bmatrix}$.

## C Proofs

### C.1 Proof of relation between DAIR-SQ and DAIR-L1 (Lemma 1)

*Proof of Lemma 1.* We proceed as follows:

$$\mathcal{R}_{\text{L1}}(z, \widetilde{z}; \theta) - \mathcal{R}_{\text{sq}}(z, \widetilde{z}; \theta) = 2\sqrt{\min\{\ell(z; \theta), \ell(\widetilde{z}; \theta)\}} \left| \sqrt{\ell(\widetilde{z}; \theta)} - \sqrt{\ell(z; \theta)} \right|,$$

We break it into two cases: if $\ell(\widetilde{z}; \theta) > \ell(z; \theta)$:

$$
\begin{aligned}
\mathcal{R}_{\text{L1}}(z, \widetilde{z}; \theta) - \mathcal{R}_{\text{sq}}(z, \widetilde{z}; \theta) &= \ell(\widetilde{z}; \theta) - \ell(z; \theta) - (\sqrt{\ell(\widetilde{z}; \theta)} - \sqrt{\ell(z; \theta)})^2 \\
&= \ell(\widetilde{z}; \theta) - \ell(z; \theta) - \ell(\widetilde{z}; \theta) - \ell(z; \theta) + 2\sqrt{\ell(\widetilde{z}; \theta)}\sqrt{\ell(z; \theta)} \\
&= -2\ell(z; \theta) + 2\sqrt{\ell(\widetilde{z}; \theta)}\sqrt{\ell(z; \theta)} \\
&= 2\sqrt{\ell(z; \theta)}(\sqrt{\ell(\widetilde{z}; \theta)} - \sqrt{\ell(z; \theta)}).
\end{aligned}
$$

If $\ell(\widetilde{z}; \theta) \le \ell(z; \theta)$:

$$
\begin{aligned}
\mathcal{R}_{\text{L1}}(z, \widetilde{z}; \theta) - \mathcal{R}_{\text{sq}}(z, \widetilde{z}; \theta) &= \ell(z; \theta) - \ell(\widetilde{z}; \theta) - (\sqrt{\ell(\widetilde{z}; \theta)} - \sqrt{\ell(z; \theta)})^2 \\
&= \ell(z; \theta) - \ell(\widetilde{z}; \theta) - \ell(\widetilde{z}; \theta) - \ell(z; \theta) + 2\sqrt{\ell(\widetilde{z}; \theta)}\sqrt{\ell(z; \theta)} \\
&= -2\ell(\widetilde{z}; \theta) + 2\sqrt{\ell(\widetilde{z}; \theta)}\sqrt{\ell(z; \theta)} \\
&= 2\sqrt{\ell(\widetilde{z}; \theta)}(\sqrt{\ell(z; \theta)} - \sqrt{\ell(\widetilde{z}; \theta)}).
\end{aligned}
$$

If we combine the two cases, we have:

$$\mathcal{R}_{\text{L1}}(z, \widetilde{z}; \theta) - \mathcal{R}_{\text{sq}}(z, \widetilde{z}; \theta) = 2\sqrt{\min\{\ell(z; \theta), \ell(\widetilde{z}; \theta)\}} \left| \sqrt{\ell(\widetilde{z}; \theta)} - \sqrt{\ell(z; \theta)} \right|.$$

$\square$

### C.2 Proof of dependence of DAIR-SQ on $\lambda$ (Proposition 3)

*Proof of Proposition 3.* We start the proof with the objective value.

$$\widehat{\mathbb{E}}_z \left\{ \frac{\ell(z; \theta_\lambda^\star) + \ell(\widetilde{z}; \theta_\lambda^\star)}{2} + \lambda \mathcal{R}_{\text{sq}}(\ell(z; \theta_\lambda^\star), \ell(\widetilde{z}; \theta_\lambda^\star)) \right\} \le \widehat{\mathbb{E}}_z \left\{ \frac{\ell(z; \widetilde{\theta}) + \ell(\widetilde{z}; \widetilde{\theta})}{2} + \lambda \mathcal{R}_{\text{sq}}(\ell(z; \widetilde{\theta}), \ell(\widetilde{z}; \widetilde{\theta})) \right\} \quad (8)$$

$$\widehat{\mathbb{E}}_z \left\{ \frac{\ell(z; \theta_\lambda^\star) + \ell(\widetilde{z}; \theta_\lambda^\star)}{2} + \lambda \mathcal{R}_{\text{sq}}(\ell(z; \theta_\lambda^\star), \ell(\widetilde{z}; \theta_\lambda^\star)) \right\} \le \widehat{\mathbb{E}}_z \left\{ \frac{\ell(z; \widetilde{\theta}) + \ell(\widetilde{z}; \widetilde{\theta})}{2} \right\} \quad (9)$$

$$\widehat{\mathbb{E}}_z \left\{ \lambda \mathcal{R}_{\text{sq}}(\ell(z; \theta_\lambda^\star), \ell(\widetilde{z}; \theta_\lambda^\star)) \right\} \le \widehat{\mathbb{E}}_z \left\{ \frac{\ell(z; \widetilde{\theta}) + \ell(\widetilde{z}; \widetilde{\theta})}{2} - \frac{\ell(z; \theta_\lambda^\star) + \ell(\widetilde{z}; \theta_\lambda^\star)}{2} \right\} \quad (10)$$

$$\widehat{\mathbb{E}}_z \left\{ \lambda \mathcal{R}_{\text{sq}}(\ell(z; \theta_\lambda^\star), \ell(\widetilde{z}; \theta_\lambda^\star)) \right\} \le \widehat{\mathbb{E}}_z \left\{ \frac{\ell(z; \widetilde{\theta}) + \ell(\widetilde{z}; \widetilde{\theta})}{2} \right\} \quad (11)$$

$$\widehat{\mathbb{E}}_z \left\{ \mathcal{R}_{\text{sq}}(\ell(z; \theta_\lambda^\star), \ell(\widetilde{z}; \theta_\lambda^\star)) \right\} \le \widehat{\mathbb{E}}_z \left\{ \frac{\ell(z; \widetilde{\theta}) + \ell(\widetilde{z}; \widetilde{\theta})}{2\lambda} \right\}. \quad (12)$$

Note equation 8 holds since $\theta_\lambda^\star$ is the minimizer of the $f_{\text{DAIR}, \mathcal{R}_{\text{sq}}, \lambda}(z, \widetilde{z}; \theta)$. equation 9 and equation 11 hold since $\mathcal{R}(\cdot)$ and $\ell(\cdot)$ are non-negative respectively. $\square$

### C.3 Proof of convergence of DAIR-SQ (Theorem 4)

We first present DAIR applied to training neural networks with Gradient Descent (GD) and followed by proof of Theorem 4.

---

**Algorithm 1** Training Neural Networks with GD

---

1: **Input:** Number of steps $T$, Training set $\mathcal{S}$, Learning Rate $\eta$, Initialized Parameter $\theta^0$
2: **for** $t = 1, 2, \ldots, T$ **do**
3:      Compute $\nabla_\theta \widehat{\mathbb{E}} f_{\mathrm{DAIR},\mathcal{R},\lambda}(z_i, \widetilde{z}_i; \theta^t)$.
4:      Set $\theta^{t+1} = \theta^t - \eta \nabla_\theta \widehat{\mathbb{E}} f_{\mathrm{DAIR},\mathcal{R},\lambda}(z_i, \widetilde{z}_i; \theta^t)$.
5: **end for**

---

**Remark 8.** *Algorithm 1 shows DAIR applied to training neural networks with GD. Note Algorithm 1 can also be extended to the stochastic setting, which is the variant we used in the experiments of Sections 3 and 4.*

*Proof of Theorem 4.* The objective function of DAIR is the following:

$$f_{\mathrm{DAIR}}(\mathbf{x}, \widetilde{\mathbf{x}}, y; \theta) = f_{\mathrm{DA\text{-}ERM}}(\mathbf{x}, \widetilde{\mathbf{x}}, y; \theta) + \lambda \left( \sqrt{\ell(\mathbf{x}, y; \theta)} - \sqrt{\ell(\widetilde{\mathbf{x}}, y; \theta)} \right)^2$$
$$= \frac{1}{2} \left( \ell(\mathbf{x}, y; \theta) + \ell(\widetilde{\mathbf{x}}, y; \theta) \right) + \lambda \left( \sqrt{\ell(\mathbf{x}, y; \theta)} - \sqrt{\ell(\widetilde{\mathbf{x}}, y; \theta)} \right)^2.$$

Substituting logistic loss function, the above equation reduces to

$$f_{\mathrm{DAIR}}(\mathbf{x}, \widetilde{\mathbf{x}}, y; \theta) = \frac{1}{2} \underbrace{\left( \log(1 + \exp(\zeta_1(\mathbf{x}, y, \theta))) + \log(1 + \exp(\zeta_2(\widetilde{\mathbf{x}}, y, \theta))) \right)}_{L(\theta) = h(\boldsymbol{\zeta}(\theta))}$$
$$+ \lambda \underbrace{\left( \sqrt{\log(1 + \exp(\zeta_1(\mathbf{x}, y, \theta)))} - \sqrt{\log(1 + \exp(\zeta_2(\widetilde{\mathbf{x}}, y, \theta)))} \right)^2}_{\mathcal{R}_{\mathrm{sq}}(\theta) = g(\boldsymbol{\zeta}(\theta))},$$

where $\boldsymbol{\zeta} = [\zeta_1(\cdot) \ \zeta_2(\cdot)]^\top$, $\zeta_1(\mathbf{x}, y, \theta) = -y\theta^\top \mathbf{x}$ and $\zeta_2(\widetilde{\mathbf{x}}, y, \theta) = -y\theta^\top \widetilde{\mathbf{x}}$. In order to obtain the convergence rate of gradient descent, we need to compute the Lipschitz constant of the gradient of $f_{\mathrm{DAIR}}$. To this end, we need to bound the Hessian of $\mathcal{R}_{\mathrm{sq}}(\theta)$. Applying chain rule to this function, we obtain:

$$\nabla_\theta^2 \mathcal{R}_{\mathrm{sq}}(\theta) = \nabla_\theta \boldsymbol{\zeta}(\theta)^\top \nabla_{\boldsymbol{\zeta}}^2 g(\boldsymbol{\zeta}) \nabla_\theta \boldsymbol{\zeta}(\theta) + \frac{\partial g}{\partial \zeta_1} \nabla_\theta^2 \zeta_1(\theta) + \frac{\partial g}{\partial \zeta_2} \nabla_\theta^2 \zeta_2(\theta) = \begin{bmatrix} -y\mathbf{x}^\top \\ -y\widetilde{\mathbf{x}}^\top \end{bmatrix}^\top \nabla_{\boldsymbol{\zeta}}^2 g(\boldsymbol{\zeta}) \begin{bmatrix} -y\mathbf{x}^\top \\ -y\widetilde{\mathbf{x}}^\top, \end{bmatrix}.$$

Where the last inequality is due to the fact that $\nabla_\theta^2 \zeta_1(\theta) = \nabla_\theta^2 \zeta_2(\theta) = 0$ since $\zeta_1(\cdot)$ and $\zeta_2(\cdot)$ are linear functions in $\theta$. Recall $g(\boldsymbol{\zeta}) = \left( \sqrt{\log(1 + \exp(\zeta_1))} - \sqrt{\log(1 + \exp(\zeta_2))} \right)^2$, which implies:

$$\left(\nabla^2_{\boldsymbol{\zeta}} g(\boldsymbol{\zeta})\right)_{11} = \frac{\exp \zeta_1 \left(-2 \log\left(\exp \zeta_1 + 1\right) \sqrt{\log\left(\exp \zeta_2 + 1\right)} + \exp \zeta_1 \sqrt{\log\left(\exp \zeta_2 + 1\right)} + 2 \log^{\frac{3}{2}}\left(\exp \zeta_1 + 1\right)\right)}{2 \left(\exp \zeta_1 + 1\right)^2 \log^{\frac{3}{2}}\left(\exp \zeta_1 + 1\right)},$$

$$\left(\nabla^2_{\boldsymbol{\zeta}} g(\boldsymbol{\zeta})\right)_{12} = \frac{\exp\left(\zeta_1 + \zeta_2\right)}{2 \left(\exp \zeta_1 + 1\right)\left(\exp \zeta_2 + 1\right)\sqrt{\log\left(\exp \zeta_1 + 1\right)}\sqrt{\log\left(\exp \zeta_2 + 1\right)}},$$

$$\left(\nabla^2_{\boldsymbol{\zeta}} g(\boldsymbol{\zeta})\right)_{21} = \frac{\exp\left(\zeta_1 + \zeta_2\right)}{2 \left(\exp \zeta_1 + 1\right)\left(\exp \zeta_2 + 1\right)\sqrt{\log\left(\exp \zeta_1 + 1\right)}\sqrt{\log\left(\exp \zeta_2 + 1\right)}},$$

$$\left(\nabla^2_{\boldsymbol{\zeta}} g(\boldsymbol{\zeta})\right)_{22} = \frac{\exp \zeta_2 \left(-2 \log\left(\exp \zeta_2 + 1\right) \sqrt{\log\left(\exp \zeta_1 + 1\right)} + \exp \zeta_2 \sqrt{\log\left(\exp \zeta_1 + 1\right)} + 2 \log^{\frac{3}{2}}\left(\exp \zeta_2 + 1\right)\right)}{2 \left(\exp \zeta_2 + 1\right)^2 \log^{\frac{3}{2}}\left(\exp \zeta_2 + 1\right)}.$$

The Lipschitz constant of the gradient is equal to the spectral norm of the Hessian. To bound that, we use the fact that the spectral norm is bounded by the Frobenius norm and hence we need to bound each individual entry of $\nabla^2_{\boldsymbol{\zeta}} g(\boldsymbol{\zeta})$. To this end, we will leverage the following inequality throughout our process:

$$1 - \frac{1}{\varpi} \leq \log \varpi \leq \varpi - 1, \quad \forall \varpi > 0. \tag{13}$$

We now bound $(\nabla^2_{\boldsymbol{\zeta}} g(\boldsymbol{\zeta}))_{11}$. Notice that, using equation 13, we have:

$$0 \leq \frac{\left(\exp \zeta_1\right)^2}{\left(\exp \zeta_1 + 1\right)^2 \log^{\frac{3}{2}}\left(\exp \zeta_1 + 1\right)} \leq \frac{\left(\exp \zeta_1\right)^2}{\left(\exp \zeta_1 + 1\right)^2 \left(1 - \frac{1}{1 + \exp \zeta_1}\right)^{\frac{3}{2}}} = \frac{\sqrt{e^{\zeta_1}}}{\sqrt{e^{\zeta_1} + 1}} \leq 1,$$

$$0 \leq \frac{\exp \zeta_1 \log\left(\exp \zeta_1 + 1\right)}{\left(\exp \zeta_1 + 1\right)^2 \log^{\frac{3}{2}}\left(\exp \zeta_1 + 1\right)} \leq \frac{\left(\exp \zeta_1\right)^2}{\left(\exp \zeta_1 + 1\right)^2 \left(1 - \frac{1}{1 + \exp \zeta_1}\right)^{\frac{3}{2}}} = \frac{\sqrt{e^{\zeta_1}}}{\sqrt{e^{\zeta_1} + 1}} \leq 1,$$

and $0 \leq \frac{\exp \zeta_1}{\left(\exp \zeta_1 + 1\right)^2} \leq \frac{1}{4}$. Putting these pieces together, we obtain $-\sqrt{\log\left(\exp \zeta_2 + 1\right)} \leq (\nabla^2_{\boldsymbol{\zeta}} g(\boldsymbol{\zeta}))_{11} \leq \frac{1}{2}\sqrt{\log\left(\exp \zeta_2 + 1\right)} + \frac{1}{8}$, which in term implies

$$(\nabla^2_{\boldsymbol{\zeta}} g(\boldsymbol{\zeta}))_{11} = \mathcal{O}(\sqrt{\mathcal{D}_x \mathcal{D}_\theta}).$$

Similarly, we can obtain $(\nabla^2_{\boldsymbol{\zeta}} g(\boldsymbol{\zeta}))_{22} = \mathcal{O}(\sqrt{\mathcal{D}_x \mathcal{D}_\theta})$. We now bound $(\nabla^2_{\boldsymbol{\zeta}} g(\boldsymbol{\zeta}))_{12}$ and $(\nabla^2_{\boldsymbol{\zeta}} g(\boldsymbol{\zeta}))_{21}$. By equation 13, we have:

$$0 \leq \frac{\exp \zeta_1}{(1 + \exp \zeta_1)\sqrt{\log(1 + \exp \zeta_1)}} \leq \frac{\exp \zeta_1}{(1 + \exp \zeta_1)\sqrt{1 - \frac{1}{1 + \exp \zeta_1}}}$$

$$= \frac{\exp \zeta_1}{\sqrt{(1 + \exp \zeta_1)^2 - (1 + \exp \zeta_1)}}$$

$$= \frac{\exp \zeta_1}{\sqrt{1 + \exp \zeta_1}\sqrt{\exp \zeta_1}} \leq 1.$$

Therefore, we have $(\nabla^2_{\boldsymbol{\zeta}} g(\boldsymbol{\zeta}))_{12} \leq \frac{1}{2}$ and $(\nabla^2_{\boldsymbol{\zeta}} g(\boldsymbol{\zeta}))_{21} \leq \frac{1}{2}$. Given the bounds of the four entries of $\nabla^2_{\boldsymbol{\zeta}} g(\boldsymbol{\zeta})$ above, we have

$$\|\nabla^2_{\boldsymbol{\zeta}} g(\boldsymbol{\zeta})\|_2 = \mathcal{O}(\sqrt{\mathcal{D}_x \mathcal{D}_\theta}).$$

Recall $\nabla_\theta^2 \mathcal{R}_{\text{sq}}(\theta) = \begin{bmatrix} -y\mathbf{x}^\top \\ -y\widetilde{\mathbf{x}}^\top \end{bmatrix}^\top \nabla_\zeta^2 g(\boldsymbol{\zeta}) \begin{bmatrix} -y\mathbf{x}^\top \\ -y\widetilde{\mathbf{x}}^\top \end{bmatrix}$ and the boundedness assumption on $\mathbf{x}$ and $\widetilde{\mathbf{x}}$, finally we have:

$$\|\nabla_\theta^2 \mathcal{R}_{\text{sq}}(\theta)\|_2 \leq \left\| \begin{bmatrix} -y\mathbf{x}^\top \\ -y\widetilde{\mathbf{x}}^\top \end{bmatrix} \right\|_2 \|\nabla_\zeta^2 g(\boldsymbol{\zeta})\|_2 \left\| \begin{bmatrix} -y\mathbf{x}^\top \\ -y\widetilde{\mathbf{x}}^\top \end{bmatrix} \right\|_2 = \mathcal{O}(\mathcal{D}_x^2 \sqrt{\mathcal{D}_x \mathcal{D}_\theta}).$$

We now find $\|\nabla_\theta^2 L(\theta)\|_2$ using similar approach. Notice that

$$\nabla_\theta^2 L(\theta) = \begin{bmatrix} -y\mathbf{x}^\top \\ -y\widetilde{\mathbf{x}}^\top \end{bmatrix}^\top \nabla_\zeta^2 h(\boldsymbol{\zeta}) \begin{bmatrix} -y\mathbf{x}^\top \\ -y\widetilde{\mathbf{x}}^\top \end{bmatrix} \quad \text{and} \quad \nabla_\zeta^2 h(\boldsymbol{\zeta}) = \begin{bmatrix} \dfrac{\exp \zeta_1}{(\exp \zeta_1 + 1)^2} & 0 \\ 0 & \dfrac{\exp \zeta_2}{(\exp \zeta_2 + 1)^2} \end{bmatrix}.$$

Thus, $\|\nabla_\zeta^2 h(\boldsymbol{\zeta})\|_2 = \mathcal{O}(1)$ and therefore $\|\nabla_\theta^2 L(\theta)\|_2 = \mathcal{O}(\mathcal{D}_x^2)$. Recall that

$$f_{\text{DAIR}}(\mathbf{x}, \widetilde{\mathbf{x}}, y; \theta) = f_{\text{DA-ERM}}(\mathbf{x}, \widetilde{\mathbf{x}}, y; \theta) + \lambda \left( \sqrt{\ell(\mathbf{x}, y; \theta)} - \sqrt{\ell(\widetilde{\mathbf{x}}, y; \theta)} \right)^2 = 2L(\theta) + \lambda \mathcal{R}_{\text{sq}}(\theta).$$

Thus, using the computed bounds $\|\nabla_\theta^2 \mathcal{R}_{\text{sq}}(\theta)\|_2 = \mathcal{O}(\mathcal{D}_x^2 \sqrt{\mathcal{D}_x \mathcal{D}_\theta})$ and $\|\nabla_\theta^2 L(\theta)\|_2 = \mathcal{O}(\mathcal{D}_x^2)$, we have

$$\|\nabla_\theta^2 f_{\text{DAIR}}(\mathbf{x}, \widetilde{\mathbf{x}}, y; \theta)\|_2 = \mathcal{O}((1 + \lambda \sqrt{\mathcal{D}_x \mathcal{D}_\theta}) \mathcal{D}_x^2).$$

Now that we have shown that the Lipschitz constant of the gradient is bounded, by classic gradient descent results (Nesterov, 2003, Theorem 1.2.4), we know that after $T$ iterations of gradient descent with stepsize $\mathcal{O}(\frac{1}{(1 + \lambda \sqrt{\mathcal{D}_x \mathcal{D}_\theta}) \mathcal{D}_x^2})$, we have

$$\|\nabla_\theta f_{\text{DAIR}, \mathcal{R}, \lambda}(\cdot)\|_2 = \mathcal{O}\left( \frac{(1 + \lambda \sqrt{\mathcal{D}_x \mathcal{D}_\theta}) \mathcal{D}_x^2}{\sqrt{T}} \right),$$

which completes the proof.

$\square$

# D   Model architecture and training parameters for MNIST experiments

We use a Convolutional Neural Network (CNN) with three convolutional layers followed by two fully connected layers. The last layer output size for Colored MNIST experiments is set to 1, and 10 for the Rotated MNIST experiments. For training we follow a two stage schedule with a learning rate of 0.005 for the first 20 epochs and a learning rate of 0.0005 for the next 20. We choose a batch size of 64 for all experiments. The architectural details and training parameters can be found in Table 8 and Table 9.

| Layer Type | Shape |
|---|---|
| Convolution + ReLU | $4 \times 4 \times 6$ |
| Max Pooling | $2 \times 2$ |
| Convolution + ReLU | $4 \times 4 \times 16$ |
| Max Pooling | $2 \times 2$ |
| Convolution + ReLU | $4 \times 4 \times 96$ |
| Fully Connected + ReLU | 64 |
| Fully Connected | $C$ |

Table 8: Model Architecture, $C = 1$ for Colored MNIST and $C = 10$ for Rotated MNIST.

| Parameter | Value | |
|---|---|---|
| Learning Rate | 0.005 | 0.0005 |
| Epochs | First 20 | Second 20 |
| Batch-size | 64 | |

Table 9: Training parameter of MNIST experiments.

We apply the proposed loss function (DAIR) on the following two datasets: Colored MNIST and Rotated MNIST. We compare the performance of DAIR with plain data augmentation, and invariant risk minimization (IRM) as a strong baseline. One crucial difference between our work and IRM is is the motivation. IRM is designed to take two examples from two different environments and learn representations that are invariant to the environment, e.g., in cases where we are aggregating multiple datasets. On the other hand, we are interested in promoting invariance when we have a single dataset. As such, we artificially generate the second environment in IRM using data augmentation. For a given example $z$, we design an augmenter $A(\cdot)$ and use it to generate additional samples that adhere to the invariance we have in mind. Hence, IRM will be applied in the same way that examples from different environments are augmenting pairs.

Our Colored MNIST is an extension of the original Colored MNIST Arjovsky et al. (2019). The label is a noisy function of both digit and color. The digit has a correlation of 0.75 with the label and a certain correlation with the label depending on the color scheme. Besides the two colors in the original dateset, we introduce fully random colored scheme to the dateset, which is the best augmenter one can think of. The three color schemes are detailed in Table 10.

Our Rotated MNIST is a variant of the original Rotated MNIST (Ghifary et al., 2015). The original dataset contains images of digits rotated $d$ degrees, where $d \in \mathcal{D} \triangleq \{0, 15, 30, 45, 60, 75\}$. Similarly, we introduce the random degree scheme here to serve as the best possible augmenter. To further exploit the potential of the proposed algorithm, we make this dataset more difficult by introducing more challenging degree scheme; The rotation schemes are summarized in Table 11.

Note all the augmented images are generated on the fly. Examples of images from some transformation schemes are shown in Figures 11 to 16.

| Scheme | $z$ | Color $\mid y = 0$ |
|--------|-----|--------------------|
| C1 | with $p = 0.8$, $z = y$ | Red |
|    | with $p = 0.2$, $z = 1 - y$ | Green |
| C2 | with $p = 0.9$, $z = y$ | Red |
|    | with $p = 0.1$, $z = 1 - y$ | Green |
| C3 | with $p = 0.1$, $z = y$ | Red |
|    | with $p = 0.9$, $z = 1 - y$ | Green |
| C4 | $z = 2$ | Random |

Table 10: Color schemes in Colored MNIST. Random color means that the value of each channel of the image is uniformly random chosen from 0 to 255.

| Scheme | Rotation |
|--------|----------|
| R1 | $0°$ |
| R2 | $90°$ |
| R3 | $0°, 180°$ |
| R4 | $90°, 270°$ |
| R5 | $[0°, 360°]$ |
| R6 | $[22.5°, 67.5°], [202.5°, 247.5°]$ |

Table 11: Rotation schemes in Rotated MNIST. $[a, b]$ means that degrees are unformly random chosen between $a$ and $b$.

| Setup Name | Train | Aug | Test | $\lambda$ |
|------------|-------|-----|------|-----------|
| Adv. Aug. | C1 | C2 | C3 | 1000 |
| Rnd. Aug. | C1 | C4 | C3 | 100 |

Table 12: Training procedure of Colored MNIST.

| Setup | Train | Aug | Test | $\lambda$ |
|-------|-------|-----|------|-----------|
| Strong Aug. | R1 | R5 | R2 | 1 |
| Weak Aug. | R4 | R6 | R3 | 10 |

Table 13: Training procedure of Rotated MNIST

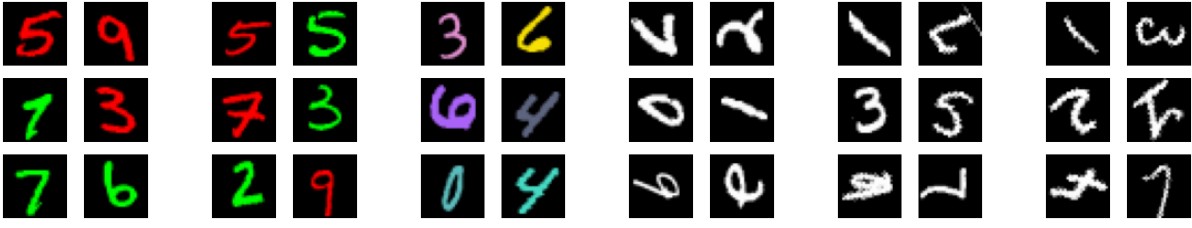

Figure 11: C2    Figure 12: C3    Figure 13: C4    Figure 14: R4    Figure 15: R5    Figure 16: R6

**Setup:** We train a model consisted of three convolutional layers and two fully connected layers with 20,000 examples. For each dataset we are defining several different schemes on how the dataset could be modified: Table 10 (Colored MNIST) and Table 11 (Rotated MNIST). Then, we define several *setups*. Each setup

is consisted of one original dataset, one augmentation dataset, and one test dataset, each of which is selected among the defined schemes. These setups are provided in Table 12 (Colored MNIST) and Table 13 (Rotated MNIST). For each setup, we train the model with the following four algorithms and compare their performances: ERM, DA-ERM, DAIR and Invariant Risk Minimization (IRM). Each experiment is repeated for 10 times; the mean and the standard derivation are reported. The value of $\lambda$ are chosen base on the validation results. Detailed architectures and training parameters can be found in Appendix D.

## D.1   Colored MNIST

We conduct two sets of experiments for this dataset: Adversarial Augmentation Setup (Table 12) follows the exact same color schemes from the original Colored MNIST Arjovsky et al. (2019). For Random Augmentation Setup, we train the model with the strongest possible augmenter: uniformly random color. The entire procedure is summarized in Table 12.

## D.2   Rotated MNIST

We start with the strongest augmenter case. One may notice that there is a chance that the augmented images bear the same rotation degrees as the testing set. To make the task more difficult, we will use R6 as the augmented test to test how the trained model generalize to entirely unseen domain. The training procedure is summarized in Table 13.

# E   Additional results on Colored MNIST & Rotated MNIST

## E.1   Colored MNIST

We show additional results on Colored MNIST and Rotated MNIST in Tables 14 and 15. Note that each algorithm has been tuned for best performance. As mentioned in Section 4.1, DAIR outperforms DA-ERM, ERM and other baseline models on classification accuracy. For accuracy consistency, we use the training scheme as the original scheme and the testing scheme as the augmentation scheme. We further compare DAIR with IRM (Arjovsky et al., 2019), DRO (Sagawa et al., 2019), and REx (Krueger et al., 2021). In doing so, we feed all original examples as one environment and all augmented examples as a second environment to these baselines. While we can see that DAIR outperforms all baselines, we caution that the comparison may not be fair in that DAIR exploits pairing information between original and augmented samples, which is not used by the other baselines.

| Algorithm | Accuracy | CM |
|---|---|---|
| ERM | $32.70 \pm 0.45$ | $77.76 \pm 1.01$ |
| DA-ERM | $40.91 \pm 0.45$ | $84.60 \pm 0.60$ |
| DAIR | $72.58 \pm 0.11$ | $99.39 \pm 0.11$ |
| IRM (Arjovsky et al., 2019) | $66.90$ | $-$ |
| DRO (Sagawa et al., 2019) | $37.40$ | $-$ |
| REx (Krueger et al., 2021) | $68.70$ | $-$ |

Table 14: Accuracy and Accuracy Consistency Metric (CM) on Colored MNIST with Adversarial Augmentation.

| Algorithm | Accuracy | CM |
|---|---|---|
| ERM | $32.70 \pm 0.45$ | $63.50 \pm 1.92$ |
| DA-ERM | $29.61 \pm 0.80$ | $88.15 \pm 0.18$ |
| DAIR | $73.10 \pm 0.12$ | $99.88 \pm 0.01$ |

Table 15: Accuracy and Accuracy Consistency Metric (CM) on Colored MNIST with Random Augmentation.

## E.2   Rotated MNIST

We report the accuracy consistency on Rotated MNIST (weak augmentation) in Table 16. The original training scheme here is Scheme R4 (Table 11), i.e., 90° and 270° rotated images, and the augmentation scheme for training is R6 (weak rotation). At test time, we test with R1 (no rotation) and we also use the augmentation scheme of 180° rotation to test the accuracy consistency metric. Note that neither the un-rotated or 180° rotated images have been observed at training time. Hence, the setup is difficult for ERM which struggles to generalize. As can be seen, since the digit 0 is "almost" circularly symmetric, ERM actually does a decent job at classifying 0, however it significantly struggles with all other digits. We see that DAIR outperforms ERM and DA-ERM by a large margin. We observe that digits 6 and 9 are challenging to get right (as one would expect for them to be difficult to tell apart). While we see $2-3\%$ drop on the consistency for digits 6 and 9 (when rotating them by 180°), the drop is smaller than expected perhaps due to the fact that the neural network learns to classify these digits based on features that are harder to get for humans.

| Digit | ERM | | DA-ERM | | DAIR | |
|---|---|---|---|---|---|---|
| | Acc. | CM | Acc. | CM | Acc. | CM |
| 0 | $86.19 \pm 01.48$ | $94.95 \pm 01.53$ | $95.61 \pm 00.66$ | $98.43 \pm 00.21$ | $98.44 \pm 00.07$ | $99.31 \pm 00.15$ |
| 1 | $00.15 \pm 00.08$ | $11.11 \pm 11.11$ | $82.79 \pm 03.38$ | $98.54 \pm 00.43$ | $96.09 \pm 00.71$ | $97.59 \pm 01.28$ |
| 2 | $29.84 \pm 00.51$ | $57.91 \pm 02.76$ | $76.68 \pm 03.54$ | $82.70 \pm 03.27$ | $86.21 \pm 00.82$ | $93.21 \pm 01.32$ |
| 3 | $00.63 \pm 00.53$ | $76.47 \pm 23.53$ | $78.84 \pm 02.60$ | $89.24 \pm 01.26$ | $86.60 \pm 02.24$ | $94.26 \pm 00.36$ |
| 4 | $01.97 \pm 00.90$ | $23.38 \pm 13.49$ | $51.09 \pm 03.30$ | $78.15 \pm 02.73$ | $79.67 \pm 01.26$ | $92.42 \pm 00.41$ |
| 5 | $05.53 \pm 00.32$ | $39.91 \pm 04.59$ | $65.02 \pm 02.42$ | $84.68 \pm 03.71$ | $83.26 \pm 02.51$ | $95.11 \pm 01.46$ |
| 6 | $00.66 \pm 00.37$ | $51.79 \pm 25.13$ | $67.43 \pm 03.82$ | $83.41 \pm 05.74$ | $84.79 \pm 01.17$ | $92.78 \pm 01.71$ |
| 7 | $16.67 \pm 02.75$ | $18.28 \pm 06.65$ | $56.29 \pm 07.26$ | $81.67 \pm 06.90$ | $78.11 \pm 02.10$ | $95.03 \pm 01.21$ |
| 8 | $10.92 \pm 05.47$ | $22.54 \pm 05.46$ | $74.50 \pm 01.10$ | $89.12 \pm 01.69$ | $90.55 \pm 01.13$ | $95.35 \pm 00.47$ |
| 9 | $17.08 \pm 07.70$ | $11.56 \pm 00.62$ | $69.54 \pm 04.18$ | $86.78 \pm 01.08$ | $80.84 \pm 01.18$ | $93.21 \pm 01.39$ |
| All | $16.85 \pm 1.08$ | $64.14 \pm 2.69$ | $71.98 \pm 1.70$ | $88.28 \pm 0.27$ | $86.57 \pm 0.55$ | $94.98 \pm 0.29$ |

Table 16: Rotated MNIST with 90° or 270° rotated original images and Weak Augmentation during training. The test scheme is un-rotated original images. Consistency metric (CM) is computed between un-roated images and ones with 180° rotation. It can be seen that CM is relatively small for 6 and 9 but the drop is smaller than expected suggesting that CNNs learn from features different from how humans perceive the digits.

# F  Practical considerations when DAIR is used in training

## F.1  Explanations for a practical sweet spot for the performance of DAIR-SQ as a function of $\lambda$

In this section, we investigate the reason why we see a sweet spot for the performance of DAIR-SQ as a function of $\lambda$. As shown in Figure 7, we see a sweet spot for $\lambda$, where the performance takes its maximum and starts to decrease for larger values of $\lambda$. There are a few explanations for this performance degradation.

1. It is observed that a large $\lambda$ requires a relatively longer time for convergence. To show empirically this is true, we added another example in Appendix F.2. Theoretically, this is in line with the classical results in the optimization literature where larger Lipschitz constants (resulting from adding a regularizer) slows down the convergence rate. Thus, as we are training all models for a certain number of epochs, we will end up with underfitting.

2. A larger $\lambda$ is more likely to guide the optimization trajectory towards a spurious poor local minimum with poor generalization performance, when the optimization trajectory is non-convex. We have experimentally verified this in Section 2.2.1 (Figure 17) as the reason for the poor performance of DAIR-L1 in Figure 7.

3. With a finite number of samples our regularizer does not necessarily lead to the best possible performance in the infinite sample setting (with weak domain shift). Hence, we might expect to observe the classical approximation-estimation tradeoff. This is especially true in real-world scenarios where one might expect that the difficulty of the example may not necessarily be preserved through data augmentation, and hence forcing the loss to be equal on both samples might be detrimental to the overall performance, which may lead to a practical sweet spot for $\lambda$.

We dig into the experiment in Figure 7 specifically and try to understand which case is the responsible for the sweet spot in Figure 7. We extend the number of training epochs from 40 to 160, and report the accuracy for $\lambda \in \{1.43, 8.85, 16.23, 100\}$.[‡] Table 17 suggests that, when we increase the number of training epochs, the sweet spot of $\lambda$ moves from 8.85 to 16.23 and in fact we can achieve an even better performing model with accuracy 89.22 as compared to the previously reported 85.89, while the performance does not change much for the smaller values of $\lambda$. We also observe a big performance boost for larger values of $\lambda$. This suggests that in this experiment the sweet spot for $\lambda$ is caused by capping the training epochs to a finite value. Having said that, we believe that we are practically interested in using DAIR with marginal computational overhead over ERM and hence we would expect to observe such sweet spot in performance in practice as $\lambda \to \infty$.

| $\lambda$ | Acc at Epoch 40 | Acc at Epoch 160 |
|---|---|---|
| 1.43 | 79.09 | 80.19 |
| 8.85 | **85.89** | 86.60 |
| 16.23 | 82.66 | **89.22** |
| 100 | 46.95 | 69.37 |

Table 17: Testing accuracy of Rotated MNIST, Weak Augmentaion. We see the accuracy increases as we extend the number of training epochs.

## F.2  Additional evidence on growing cost of training with the regularization strength

We also provide further evidence for the growing cost of training with $\lambda$ on a toy problem where we can reliably measure the gradient norm and ensure convergence. We study the following simple binary classification problem with logistic regression, which mirrors the MNIST experiments: at the training time the input is $\mathbf{x}_{\text{train}} = (x, s = 2y - 1 + t_1)$ and the label $y$, i.e., $z_{\text{train}} = (\mathbf{x}_{\text{train}}, y)$. Here, $x \sim \mathcal{N}(0, \sigma_x^2)$, and $P(y = 1|x) = \frac{1}{1+e^{-x}}$, where $t_1$ is independent of $x$ and $t_1 \sim \mathcal{N}(0, \sigma_1^2)$. In this example, we intentionally provide feature $s$ which is highly correlated with the label during training. Again, clearly, $w^\star = (1, 0)^\top$, but

---

[‡]Note these values comes from $\log_{10}$ sweeping of $\lambda$.

$w_{\text{ERM}}^{\star}$ will converge to $(0, 1)^{\top}$ due to the overfitting to the spurious feature. We introduce an augmenter which generates the augmented example such as $\mathbf{x}_{\text{aug}} = (x, s = 2y - 1 + t_1 + t_2)$ where $t_2 \sim \mathcal{N}(0, \sigma_2^2)$. We use this data augmenter for DAIR training and test on $\mathbf{x}_{\text{test}} = (x, s = 1 - 2y)$. We summarize the steps need for convergences and the testing accuracy in Table 18 as well. We can find that the required number of iteration to convergence increases as $\lambda$ increase.

For this tiny toy example, there is a factor of 10x increase in the required number of iterations when $\lambda$ is chosen to be 10,000 as opposed to 0.5. Note that this is using ADAM and the gap is significantly larger if we use vanilla gradient descent; as we were not able to even converge in $10^8$ steps. This is provided as further evidence for the practical sweet spot for DAIR as $\lambda \to \infty$.

| $\lambda$ | Iterations to Converge |
|---|---|
| 0.5 | $81.35 \pm 6.07$ |
| 1 | $91.05 \pm 2.53$ |
| 2 | $89.10 \pm 2.41$ |
| 5 | $101.65 \pm 2.87$ |
| 10 | $107.70 \pm 5.77$ |
| 100 | $151.75 \pm 4.28$ |
| 1,000 | $195.85 \pm 4.54$ |
| 10,000 | $802.60 \pm 7.58$ |

Table 18: Iteration needed for the logistic model to converge with different $\lambda$. The model is converged when the $\mathcal{L}_2$ norm of the gradient is less than $10^{-7}$.

### F.3 Compare DAIR-SQ and DAIR-L1 on avoidance of spurious local min

We empirically verify this hypothesis on Colored MNIST with Adversarial Augmentation. Figure 17 depicts the classification loss and regularization of the first 10 and last 140 iterations. One observes that at the beginning of training, regularization term of DAIR-SQ impacts the training dynamics less while DAIR-L1 starts optimizing the regularizer right away, which dominates the entire training procedure and therefore leads the model to a poor local minimum. The left panel of Figure 17 confirms that the classification loss of DAIR-L1 remains large and unchanged (that of a random classifier).

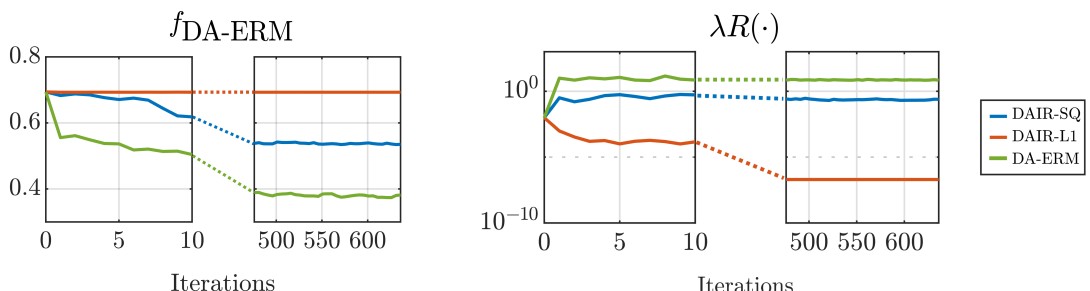

Figure 17: Training DA-ERM loss and (DAIR-SQ) for first 10 and last 140 iterations on Colored MNIST with Adv Aug for DAIR ($\lambda = 100$). The regularizer loss on DA-ERM grows large as it is uncontrolled. DAIR-L1 is optimizing an L1 regularizer, but for unified illustration we evaluate it using (DAIR-SQ).

This same property of DAIR-SQ also weakens the regularizer on training samples with high losses at the later stages of training. These samples are likely noisy, which makes DAIR-SQ more robust to noisy samples, as we already observed in Section 4.2.

For the experiments in Sections 3 and 4, we tune $\lambda$ in DAIR regularizer on top of the ERM baselines. In other words, in all experiments we use the same set of hyper-parameters including step-size, batch-size, etc., from literature and only tune $\lambda$ on the validation set.

# G    The impact of partial augmentation

We explore the impact of partial augmentation, where we only augment a certain fraction of the training samples. The experiment revisits noiseless Rotated MNIST with weak rotation data augmentation and Colored MNIST with Adversarial augmentation. This experiment emulates situations where an augmentation function is only applicable to certain examples or where augmentation is expensive and we would like to decrease the augmentation cost.

In Figure 18, we report the experiment results for DA-ERM and DAIR-SQ by applying augmentation only {10%, 20%, 30%, 50%, 100%} of the training samples, averaged on three runs. In Rotated MNIST experiment, as can be seen, DAIR-SQ with augmentation on only 20-30% of the samples performs similar to full augmentation. On the other hand, DA-ERM is more sensitive to partial augmentation and is subject to a steeper performance drop. This could be viewed as further evidence that DAIR-SQ could reach its best performance using weak augmenter functions. It is also noteworthy that in this example, DAIR-SQ with only 10% partial augmentation still outperforms DA-ERM with 100% augmentation. One can draw similar conclustion in the Colored MNIST experiment as only 10% augmentation gives comparable performance to full augmentation.

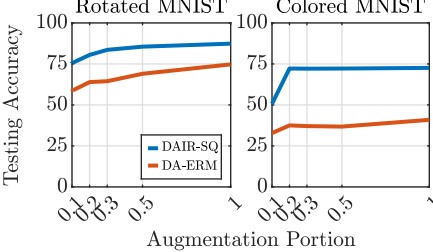

Figure 18: Test accuracy vs fraction of augmented samples on Rotated MNIST.

## H  Details on neural task-oriented dialog modeling

We provide details on the benchmark that we used in this experiment. Qian et al. (2021) proposed a new test set for MultiWOZ 2.2, called MultiWOZ 2.2 with SGD entities, where named entities are replaced with those from Schema Guided Dialog dataset (Rastogi et al., 2020) and showed that SimpleTOD (Hosseini-Asl et al., 2020) endures more than 8% performance drop on the new test set. Examples from the dataset are shown in Table 20. To address this problem, we define a new data augmentation scheme for DAIR and DA-ERM by replacing the named entities from the MultiWOZ 2.2 training set with randomly scrambled versions of the named entities. For example, "warkworth house" could be turned into "easrtokow hhrwu" (see Table 20). In all of our experiments, we utilize the SimpleTOD model (Hosseini-Asl et al., 2020) and we apply DAIR to enforce invariance between the named entities in the training examples and the scrambled entities from their corresponding augmented samples. The model is trained with ParlAI (Miller et al., 2017) fine-tuned with the pre-trained BART (Lewis et al., 2019). Training hyper-parameters can be found in Table 19. The optimal $\lambda$ was tuned by grid search in $\{0.01, 0.1, 0.3, 0.5, 0.7, 0.9, 0.99, 1.0\}$.

| Parameter | Value |
|---|---|
| $\lambda$ | 0.5 |
| Epochs | 4 |
| Batchsize | 6 |
| Optimizer | AdamW |
| Learning rate | $10^{-5}$ |

Table 19: Hyper-parameters used in training SimpleTOD.

| User: | can you help me book a reservation at the warkworth house hotel? | User: | can you help me book a reservation at the easrtokow hhrwu hotel? | User: | can you help me book a reservation at the clarion inn & suites atlanta downtown hotel? |
|---|---|---|---|---|---|
| Agent: | yes i could! how many people are staying, and what days would fyou like to stay? | Agent: | yes i could! how many people are staying, and what days would fyou like to stay? | Agent: | yes i could! how many people are staying, and what days would fyou like to stay? |
| User: | it's just for me, and i'll be staying for three nights starting from tuesday. | User: | it's just for me, and i'll be staying for three nights starting from tuesday. | User: | it's just for me, and i'll be staying for three nights starting from tuesday. |
| DS: | *hotel-bookday:* tuesday *hotel-bookpeople:* 1 *hotel-bookstay:* 3 *hotel-name:* warkworth house | DS: | *hotel-bookday:* tuesday *hotel-bookpeople:* 1 *hotel-bookstay:* 3 *hotel-name:* easrtokow hhrwu | DS: | *hotel-bookday:* tuesday *hotel-bookpeople:* 1 *hotel-bookstay:* 3 *hotel-name:* clarion inn & suites atlanta downtown |

Table 20: Left: sample from the original MultiWOZ dataset. Middle: augmented sample generated by scrambling. Right: synthetic sample with name entities from SGD. Comparing left and the middle example, we are generating new named entities (marked in red) by scrambling. Comparing left and the right example, the only difference is the named entity from different dataset, which is marked in red. Note that the SGD named entities are not exposed to the model during training. Only the original named entities and scrambled named entities from MultiWOZ are used during training.

# I   Setup and additional results for Visual Question Answering

All the approaches included in this paper use the original VQA v2 train split for training, along with the IV-VQA and CV-VQA train splits for augmentation in the DAIR and DA-ERM(Agarwal et al., 2020) settings. The ERM setup (Kazemi & Elqursh, 2017), represents a vanilla SAAA model trained on the VQA v2 train split. For the data augmentation methods, if an image from VQA v2 contains multiple edited versions in IV-VQA/CV-VQA, we randomly select one of them to serve as an augmented sample during training. We modify the official code released by Agarwal et al. (2020) to suit our formulation. All the methods are trained for 40 epochs with a learning rate of 0.001 and a batch size of 48. The baseline approaches that we compare against are trained and evaluated by us, using the same training setup as DAIR.

| $\lambda$ | VQA val (%) | CM | Predictions flipped (%) | pos $\rightarrow$ neg (%) | neg $\rightarrow$ pos (%) | neg $\rightarrow$ neg (%) |
|---|---|---|---|---|---|---|
| 0.72 | **64.89** | 95.89 | 6.67 | 2.64 | 2.38 | 1.65 |
| 1 | 64.75 | 96.06 | 6.33 | 2.54 | 2.22 | 1.57 |
| 1.68 | 63.90 | 96.19 | 5.78 | 2.20 | 1.95 | 1.64 |
| 2.68 | 62.51 | 96.63 | 5.23 | 1.88 | 1.86 | 1.49 |
| 5.18 | 60.03 | 97.22 | 4.45 | 1.63 | 1.59 | 1.22 |
| 10 | 57.70 | **97.67** | **3.91** | **1.33** | **1.37** | **1.21** |

Table 21: Accuracy-Consistency Tradeoff on VQA v2 val and IV-VQA test set controlled by $\lambda$

Table 21 indicates a tradeoff between the accuracy on the VQA v2 val set and the consistency metrics. The optimal $\lambda$ value is determined by grid search over a uniformly chosen set of size 8 in log space $[10^{-1}, 10]$ with the corresponding performance on the validation set. As the $\lambda$ value increases, the consistency between the predictions increases, while the accuracy on original examples decreases. For instance, A $\lambda$ value of 10 strongly boosts consistency thus lowering the 'Predictions flipped' percentage to only 3.91% but sacrifices the classification accuracy causing it to drop to 57.7%.

## J    Additional details on robust regression

For the robust regression task, we determine the optimal $\lambda$ by performing a grid search over a uniformly chosen set of size 5 in log space $[1, 10^4]$ and the best performing $\lambda$ on validation set is used for reporting the results on the test set. We set the learning rate to 0.01 for all these experiments. Following the convention from Li et al. (2021), we set the tilting factor 't' to -2 for all experiments that use the TERM objective.

# K    Details on training robust neural networks

## K.1    Setups for the main results in Section 4.4

For all algorithms reported in Table 5, we use Pre-Activation ResNet-18 (He et al., 2016), with a last-layer output size of 10 as the classification model and their original hyper-parameters. For training the DAIR model, the adversarial examples are generated by $\mathcal{L}_\infty$ based PGD attack with 11 iterations, $\varepsilon$ (attack strength) set to 8/255 and attack step size to 2/255. We train the model for 120 epochs with initial step size 0.0001 and uses CosineAnnealing scheduler. We evaluate all the models against the standard FGSM attack and PGD attack with 20 iterations of same perturbation sizes. The optimal $\lambda$ by performing a grid search over a uniformly chosen set in log space $[10^{-1}, 10^2]$ with 10 points.

## K.2    Additional results with new baselines

We also compare DAIR with some recent new baselines such Tack et al. (2021), which utilizes Jensen-Shannon consistency regularization on the features. To be detailed, a pair of images with attacks are fed into the model and the Jensen-Shannon distance between the resulting output logits are computed afterward as the regularizer. The regularizer then is added to existing algorithms such as (Madry et al., 2018), Zhang et al. (2019) to boost their performances. The results are summarized in Table 22. It can be seen that DAIR is also comparable with new baseline. It worth mentioning that the performance of DAIR is better in Table 22 than in Table 5. The reason is that the training and the tuning setups are different. We follow the exact setup of Tack et al. (2021) and obtain the results in this subsection.

| Method | Clean | PGD-20 |
|---|---|---|
| ERM (Madry et al., 2018) | 84.57 (83.43) | 45.04 (52.82) |
| MART (Wang et al., 2019) | 82.63 (77.00) | 51.12 (54.83) |
| TRADES (Zhang et al., 2019) | 82.87 (82.13) | 50.95 (53.98) |
| JS Consistency (Tack et al., 2021) | 86.45 (85.25) | 56.51 (57.53) |
| DAIR-SQ | 86.16 (85.24) | 56.68 (57.22) |

Table 22: DAIR vs Tack et al. (2021). Accuracies in the parenthesis are from models tuned for PGD-20 while accuracies to the left of the parenthesis are from models tuned for clean images.

## L    Details on training ImageNet-9

We use ResNet-50 provided by Torchvision but replace the last layer with 9 outputs. We train the model for 175 epochs with batchsize 128, initial learning rate of 0.1 and decay of 0.1 at 30, 70, 110, 150 epochs.

