# OpenReview forum: "Robustness through Data Augmentation Loss Consistency"
_TMLR — Accepted by TMLR_

### Review · Reviewer_TWyi · 2022-08-20

**Summary Of Contributions:**

This paper presents a method that adds to a normal training with data augmentation a consistency loss that compares the augmented sample with the original one.
The main contribution of the paper is to propose a consistency loss computed on the losses between the original and the augmented sample. This enables the method to work also on problems in which the data augmentation could be covariant, i.e. the label of the augmented sample can vary with the augmentation.


**Broader Impact Concerns:**

I am happy with the broader impact.

**Requested Changes:**

- I am not fully convinced that DAIR is better that consistency loss on logits for invariant DA. Can the authors find more examples in which DAIR is better that consistency on logits?

- In the contributions, at the end of sec.1 the authors mention that DAIR can also reduce the computational cost compared to KL on logits. Can you explain why and how? I did not find a clear example/results in the manuscript

- I think sec. 2.2 is not really adding much to the understanding of the method and the example try to give some additional capability to DAIR, which is not really there in my opinion. Thus, I would remove it.

- I do not fully understand the structure of the paper. I would like the authors explain more about this structure in which experiments are mixed with some theoretical analysis in sec. 2.



**Strengths And Weaknesses:**

\+ The paper is well written and easy to follow although I did not fully understand the choice of presenting some experiments in sec.2 and others in sec.3

\+ It is appealing to have a consistency loss that can work also for the case of covariant data augmentation. I wonder if someone already tried the same approach for unsupervised pre-training

\+ I like that the authors presented some more theoretical analysis of the proposed approach

\- Data augmentation is simpler and in many cases the difference in performance is small. Additionally, an often used scheme for data augmentation considers only the augmented samples and not the original one. In this case, adding the consistency loss would increase also the computational cost of the training as it would require the evaluate each sample only once while consistency loss would require two evaluations per sample.

\- Section 2.2 does not add much to the understanding of the proposed innovation. The proposed task is a toy example and the only reason that DAIR works better than DA is that it has a lambda value that allows the method to tune the importance of the augmentation in the final loss. Also a more classic costinsency loss based on KL on logits with a large lambda would make the job. Finally even DA in which the augmented sample loss has higher weight than the original samples would make the job. Thus, in my understanding this toy example does not bring anything to the method and it should be removed.

\- In most of the cases the performance of DAIR and KL on logits is very similar for invariant tasks. In my understanding enforcing the logits to be similar is like imposing a stronger constraint than imposing the losses to be similar. This might be the reason why for high noise (in Fig. 4) DAIR performs better than KL on logits.

Minor comments:
- Why Data augmented invariant regularisation? Finally the main contribution is the ability to deal with covariant augmentations.
- Why the authors presented DA as the combination of original samples and augmented samples? This is not what it is done in many cases.

---

> ### Author Response · Authors · 2022-08-24
> **Point-by-point response and revisions will be posted soon**
>
> Thank you for your insightful questions/comments. We are preparing a point-by-point response along with revisions to address your comments that we will share in the next few days.

---

> ### Author Response · Authors · 2022-09-18
> **Response to Reviewer TWyi (Part 2 / 2)**
>
> > In the contributions, at the end of sec.1 the authors mention that DAIR can also reduce the computational cost compared to KL on logits. Can you explain why and how? I did not find a clear example/results in the manuscript
>
> We have dropped this claim in the revision. Here is our original thought: If we only consider the loss itself, DAIR saves one pass over logits compared with KL. This time saving depends on the dimension of the logit which is the number of categories in classification or the vocab size in generative models, which can be significant if the output vocabulary size is large (e.g., ~50K in the dialog state tracking experiment) However we agree that the training might be dominated by back-prop and the time saved by DAIR may still be  marginal.  We tried to reproduce the time saved by DAIR compared to KL in this experiment but due to the complexity of the setup, it seems too complicated to do so, and decided to drop the claim.
> > I think sec. 2.2 is not really adding much to the understanding of the method and the example try to give some additional capability to DAIR, which is not really there in my opinion. Thus, I would remove it.
>
> We already gave a detailed answer to this question which we repeat here for completeness.
>
> The goal of Appendix A (Previously Section 2.2) was to start with a simple example which helped explain DAIR, which and was also amenable to analysis, as appreciated by **Reviewer TWyi**. A few points about this toy example are in order:  (1) This is a regression task and hence KL is not directly applicable unless we perform binning or use a kernel. (2) We disagree that weighted DA will do the job in this example. We also redid the experiment where we only trained on augmented samples, which we refer to as DA-ERM*-2. We observed that DA-ERM*-2 still heavily relies on the spurious feature with $w_\text{DA-ERM-2}=[0.60, 0.78]^\top$ at convergence (now added to Figure 10 in Appendix A).  Having said that, we agree with you that it is best to use a covariant toy example in this section to explain DAIR. As such, we decided to move this toy example to the appendix and replace it with a toy setup involving Rotated 6 & 9, which is a covariant example in which we can clearly show DAIR outperforms DA-ERM and KL feature consistency..
>
>
> > I do not fully understand the structure of the paper. I would like the authors explain more about this structure in which experiments are mixed with some theoretical analysis in sec. 2.
>
> With the new toy example, we now found a better way to structure our paper. After this restructure, Section 2 only talks about the theoretical results in addition to the rotated 6&9 experiment and all numerical results are in Section 3 (covariant experiments) and section 4 (invariant experiments). We intended to show that DAIR-SQ is easier to tune (and significantly outperforms DAIR-L1) and more robust in terms of choosing $\lambda$ compared with other consistency regularizations. We have now moved Section 2.3 to Section 4 (invariant experiments) in our revision.
>
> > Why Data augmented invariant regularisation? Finally the main contribution is the ability to deal with covariant augmentations.
>
> Thanks for this feedback. We have now changed the name of the proposed method to Data augmented *loss* invariant regularization (DAIR), where we wish to keep the acronym unchanged.  We would like to emphasize that DAIR is designed to have universal capability of both invariant and covariant tasks. We hope the reviewer finds this new name to remove any confusion. > Why the authors presented DA as the combination of original samples and augmented samples? This is not what it is done in many cases.
>
> We started with the formulation $\gamma L_\mbox{org} + (1-\gamma) L_\mbox{aug} + \lambda R$ in which $\gamma$ controls the weight of the original example and the augmented example. We empirically observed that $\gamma$ does not affect the performance of DAIR in toy examples and MNIST experiments. Therefore for simplicity and the ease of tuning, we fix $\gamma =0.5$ in our formulation (we added footnote 1 on page 3 to that effect). Please also notice that the value of $\gamma$ does not really affect the computational cost for DAIR (or other consistency regularizers)  as computing the regularizer needs the pair to be fed into the model anyway.
>
> We again thank the reviewers for their feedback. Please let us know if you have any further questions and comments.
>
> [1] Zhang H, Yu Y, Jiao J, Xing E, El Ghaoui L, Jordan M. Theoretically principled trade-off between robustness and accuracy. InInternational conference on machine learning 2019 May 24 (pp. 7472-7482). PMLR.
>
> [2] Li Z, Liu L, Dong C, Shang J. Overfitting or Underfitting? Understand Robustness Drop in Adversarial Training. arXiv preprint arXiv:2010.08034.

---

> ### Author Response · Authors · 2022-09-18
> **Response to Reviewer TWyi (Part 1 / 2)**
>
> Thank you for your detailed and constructive feedback. We are glad that you find our paper well written and easy to follow. We are also pleased that you are happy with our theoretical analysis of the proposed approach. We provide point-by-point responses to your questions and comments below:
>
> > Data augmentation is simpler and in many cases the difference in performance is small. Additionally, an often used scheme for data augmentation considers only the augmented samples and not the original one. In this case, adding the consistency loss would increase also the computational cost of the training as it would require the evaluate each sample only once while consistency loss would require two evaluations per sample.
>
> We agree with the reviewer that the computational cost is doubled compared with the case that only augmented samples are considered for training. This is a common cost for any approach involving consistency regularization, where both the original and augmented examples need evaluation. The DAIR regularizer itself adds negligible computational cost on top of computing gradients for those examples. Having said that we added a remark to that effect in  the limitations and broader impact section.
>
> We disagree that data augmentation is always sufficient. We show several examples where big boosts are associated with DAIR compared to DA-ERM. In particular, in new *Rotated 6&9* toy example (Section 2.1), we observe that DA-ERM hurts performance compared to ERM whereas DAIR still offers a 5+% boost over ERM test accuracy.
>
> > Section 2.2 does not add much to the understanding of the proposed innovation. The proposed task is a toy example and the only reason that DAIR works better than DA is that it has a lambda value that allows the method to tune the importance of the augmentation in the final loss. Also a more classic costinsency loss based on KL on logits with a large lambda would make the job. Finally even DA in which the augmented sample loss has higher weight than the original samples would make the job. Thus, in my understanding this toy example does not bring anything to the method and it should be removed.
>
> The goal of Appendix A (Previously Section 2.2) was to start with a simple example to help explain DAIR, which was also amenable to analysis, as appreciated by **Reviewer TWyi**. A few points about this toy example are in order:  (1) This is a regression task and hence KL is not directly applicable unless we perform binning or use a kernel. (2) We disagree that weighted DA will do the job in this example. We also redid the experiment where we only trained on augmented samples, which we refer to as DA-ERM*. We observed that DA-ERM* still heavily relies on the spurious feature with $w_\text{DA-ERM-2}=[0.60, 0.78]^\top$ at convergence (now added to Figure 10 in Appendix A).  Having said that, we agree with you that it is best to use a covariant toy example in this section to explain DAIR. As such, we decided to move this toy example to the appendix and replace it with a toy setup involving Rotated 6 & 9, which is a covariant example in which we can clearly show DAIR outperforms DA-ERM and KL feature consistency.
>
>
>
> > In most of the cases the performance of DAIR and KL on logits is very similar for invariant tasks. In my understanding enforcing the logits to be similar is like imposing a stronger constraint than imposing the losses to be similar. This might be the reason why for high noise (in Fig. 4) DAIR performs better than KL on logits.
>
> We thank the reviewer for this comment, and agree with the reviewer that in general DAIR is similar to and may not outperform existing consistency regularizations under invariant scenarios. Thanks also for the interesting hypothesis about the noise tolerance. That motivated us to perform several other experiments with label/feature noise where we did not observe a clear trend of better performance of DAIR with respect to KL consistency for moderate values of noise.
>
>
> > I am not fully convinced that DAIR is better that consistency loss on logits for invariant DA. Can the authors find more examples in which DAIR is better that consistency on logits?
>
> We thank the reviewer for pointing this fact out. We now clarify throughout the paper that our claim is that *DAIR is competitive with other consistency regularizers on invariant data augmentation, whereas it is the only form of consistency regularization that is also applicable to covariant data augmentation.*

---

### Review · Reviewer_P15o · 2022-08-20

**Summary Of Contributions:**

This paper proposes data augmented invariant regularization (DAIR) to constrain the consistency between different augmentations of the same sample from a loss level perspective. Besides, an interesting variant of DAIR-SQ as well as some theoretical analyses are also proposed. Extensive experiments have been conducted to verify the proposed method.

**Broader Impact Concerns:**

The authors have provided this part in the paper, and there is no obvious ethical implication.

**Requested Changes:**

Please see the above weaknesses.

**Strengths And Weaknesses:**

*Strengths
1. The paper is well organized, which is easy to follow.
2. The experimental results are somewhat good compared with the baselines.
3. The idea of DAIR-SQ is interesting.


*Weaknesses
1. The contribution of this paper is somewhat incremental and limited. The authors claim that “However, regularization at loss level has been relatively unexplored”.  The reviewer cannot agree with this point. In fact, the consistency regularization no matter at feature level nor loss level has been widely used in different fields, such as semi-supervised learning, knowledge distillation and adversarial training [1][2][3]. There is no essence different from DAIR with the existing works.
2. Because the covariant data augmentation will make the label to be somewhat changed, why the proposed DAIR could deal with such a kind of data augmentation?
3. Although DAIR-SQ is interesting and indeed effective, the motivation or inspiration of proposing DAIR-SQ is not clear.
4. In the experimental parts, the authors only use the name of DAIR. Is it in fact DAIR-SQ?
5. What’s the meaning of “KL feature consistency” in Table 1? How about using KL in the loss level like KL in Figures 4-6?
6. The authors only compare DAIR with the basic DA-ERM in the experimental parts, making the experimental results less convincing. The reviewer recommend the authors to compare DAIR with closely related works especially the regularization terms used in Figures 4-6.


[1] Adversarial Logit Pairing. NeurIPS 2018.

[2] Virtual Adversarial Training: A Regularization Method for Supervised and Semi-Supervised Learning. TPAMI 2019.

[3] Theoretically Principled Trade-off between Robustness and Accuracy. ICML 2019.

---

> ### Author Response · Authors · 2022-08-22
> **Response to Reviewer P15o (Part 2 / 2)**
>
> Below is the second part of our response to your comments:
>
> > What’s the meaning of “KL feature consistency” in Table 1? How about using KL in the loss level like KL in Figures 4-6?
> KL feature consistency in Tables 1 and 2 is the KL consistency regularization at the last layer (after softmax output), which was used in Figs 7-9 (Previously Figs 4-6). KL cannot be applied at the loss level (since the loss is a single scalar and KL is defined for probability vectors). We have now added a formal definition for KL feature consistency in section 2.1 to remove any confusion. As can be seen, since the nature of data augmentation is covariant in these experiments, KL feature consistency not only does not help, but also hurts performance.
> > The authors only compare DAIR with the basic DA-ERM in the experimental parts, making the experimental results less convincing. The reviewer recommend the authors to compare DAIR with closely related works especially the regularization terms used in Figures 4-6.
>
> We compare with several state-of-the-art references for specific tasks, e.g., TRADES [3] and JS Consistency (Tack et al. 2021) for adversarial attacks in Tables 4 and 21 (Previously Tables 4 and 20), KL Consistency in Tables 1 and 2, and (Xiao et al. 2020) in Table 6 (Previously Table 5). KL consistency is not directly applicable to the regression task Table 5 (Previously Table 3) due to the uncountable nature of the output. Please let us know what references you believe are missing, and we are happy to add them to the comparison mix.
> Having said that, we believe the conclusion of this work will be unchanged even if we add more baselines: *Loss level consistency regularization (DAIR-SQ) is competitive with state-of-the-art consistency regularization techniques for invariant data augmentation, while additionally it can be successfully applied to covariant data augmentation and set new state-of-the-art results, which is a case that feature-level consistency (such KL regularization at the last layer) cannot handle.*
>
> We again thank the reviewers for their feedback. Please let us know if you have any further questions and comments.

---

> ### Author Response · Authors · 2022-09-16
> **Response to Reviewer P15o (Part 1 / 2)**
>
> Thank you for your feedback and for recognizing the effectiveness of DAIR-SQ through extensive experiments, the interestingness of the DAIR-SQ variant of loss consistency, and the theory. Below we respond to the reviewers’ points on the weaknesses of the paper.
>  > The contribution of this paper is somewhat incremental and limited. The authors claim that “However, regularization at loss level has been relatively unexplored”. The reviewer cannot agree with this point. In fact, the consistency regularization no matter at feature level nor loss level has been widely used in different fields, such as semi-supervised learning, knowledge distillation and adversarial training [1][2][3]. There is no essence different from DAIR with the existing works.
> [1] Adversarial Logit Pairing. NeurIPS 2018.
> [2] Virtual Adversarial Training: A Regularization Method for Supervised and Semi-Supervised Learning. TPAMI 2019.
> [3] Theoretically Principled Trade-off between Robustness and Accuracy. ICML 2019.
>
> We thank the reviewer for pointing out these references on consistency regularization. We indeed cited several consistency regularization papers (including [1] and [3]) in the introduction (See the paragraph that starts with **Consistency regularization**). We have now also included [2] in this section. In fact, we compare with TRADES [3] on adversarial attacks benchmarks –See Section 4.3, Table 4 (Previously Section 3.2.2, Table 4) and Appendix K.2, Table 21 (Previously Appendix I.2, Table 20). [1]-[3] are consistency regularization at the feature level (e.g., last layer before loss), and hence are not applicable to covariant data augmentation, such as the experiments in Section 3 (Previously Section 3.1). See also the newly added toy example with *Rotated 6&9* in Section 2.1. We discuss this more in the response to the next question.
> > Because the covariant data augmentation will make the label to be somewhat changed, why the proposed DAIR could deal with such a kind of data augmentation?
> Indeed, this is the major contribution of this work. As you correctly pointed out, prior feature consistency approaches cannot be applied to this setting, while DAIR is applicable. We have explained in Section 2.1 why loss-level regularization is still meaningful whereas feature consistency, such as KL consitency regularization at the last layer, is not in this case. In short, loss consistency enforces the new augmented (feature, label) pair to be as likely as the original (feature, label) pair, regardless of whether the augmented label is the same as the original label. Please also see the newly added toy example with *Rotated 6&9* in Section 2.1, which provides further empirical evidence to that effect.
> > Although DAIR-SQ is interesting and indeed effective, the motivation or inspiration of proposing DAIR-SQ is not clear.
>
> The motivation for DAIR-SQ as opposed to KL feature consistency is presented in Section 2.1 (for example see the newly added *Rotated 6&9* example). Here we show that DAIR-SQ is the only method that consistently outperforms ERM.
> The motivation for DAIR-SQ as opposed to simpler variants (such as DAIR-L1) is presented in Section 2.2 (and also numerical evidence in Section 2.1). In short, the motivation is to stabilize training and tuning of the regularizer for broad applicability. Due to the particular shape of DAIR-SQ landscape, DAIR-SQ does not activate at the beginning of training. However, as the training proceeds, the regulairzer starts to force the network to become loss-invariant on the augmented samples (see the discussion after Lemma 1). We believe that the broad nature of experiments in Sections 3 specifically, and also Section 4, confirms this fact empirically as well.
>
> > In the experimental parts, the authors only use the name of DAIR. Is it in fact DAIR-SQ?
>
> All experiments in Section 3 use the DAIR-SQ variant of DAIR (unless otherwise stated explicitly). We made that clear in the beginning of Section 3. Thank you for pointing this out!

---

### Review · Reviewer_mkiw · 2022-08-23

**Summary Of Contributions:**

This work proposed a new loss function for deep learning termed DAIR. For input z and its augmented input z1, the DAIR enforces the loss-level consistency between the two input. Proposition 1 proves that even in infinite data regime, the proposed DAIR is able to outperform naive linear regression methods when the input features depends on the regression target. The authors validate the performance of DAIR on multiple datasets, including computer vision tasks and NLP tasks.


**Requested Changes:**

* More analysis on why DAIR is better than feature-level consistency methods, especially, Contrastive learning.
* Numerical experiments of comparing DAIR to CL methods, such as MOCO for example, on ImageNet. Consider using ResNet-50 and ViT as backbone.
* NLP experiments for DIAR v.s. CL are suggested as well. However, I am not an NLP expert so cannot give detailed suggestions.

**Strengths And Weaknesses:**

[Strengths]

* The paper is well written and easy to follow. All contents are well organized and well presented.
* The toy example givein in Sec 2.2 is interesting and insightful.
* Lots of experiments on multiple datasets with good diversity.

[Weakness]

One major concern is how to compare DAIR to Contrastive Learning (CL). The authors claimed that the loss-level consistency (as in DAIR) is better than feature-level consistency (as in CL). However, this key argument is not well supported by the theoretical analysis and the numerical experiments:
* From theoretical side, it is unclear why DAIR is better than the popular CL approach. The toy example in Sec 2.2 did not show that CL-based method is also guarantee to fail.
* From empirical side, there is no experiment to compare DAIR to CL on large-scale dataset and on large models. For example, it is better to compare DAIR to MOCO on ImageNet-1k.

It is unclear what Prop. 4 indicates. For example, how can we choose the $\lambda$ in practice according to Prop. 4?

Theorem 5 seems trivial. It follows the classical online learning bound for excess risk. Can authors explain why Theorem 5 is emphasized here?

---

> ### Author Response · Authors · 2022-08-24
> **DAIR vs Contrastive Learning**
>
> Thanks for your insightful review and feedback on our paper! We would like to clarify a few major points before we provide a point-by-point response along with a revision in the next few days.
>
> * **DAIR is the only form of consistency regularization that is applicable to covariant data augmentation.** The main contribution of this work is to propose a new form of consistency regularization, called DAIR, that is applicable to *covariant* data augmentation. In this case, feature consistency techniques would not be applicable; we have provided justification for the fact that standard feature consistency regularization does not apply to *covariant* data augmentation in Section 2, and empirical evidence for it in Section 3.1. This is the main use case where DAIR is needed!
>
> * **DAIR can also be applied to invariant data augmentation, resulting in competitive performance with existing techniques.** We have also compared DAIR against other forms of feature consistency regularization, and have found it to be competitive with other forms of feature consistency on several benchmarks (from Colored/Rotated MNIST to adversarial robustness). This shows that this seemingly weaker form of consistency regularization is *also* effective/competitive for invariant data augmentation but it is not necessarily the best approach, and we will make sure we do not come off as claiming that. This is more of a sanity check and a positive additional benefit of DAIR.
>
> * **Data augmentation is used far beyond contrastive learning.** Contrastive learning is just one framework where data augmentation is used/needed. On the other hand, there are a ton of practical use cases of data augmentation, which is virtually solving any practical problem using ML. In practice, there are numerous cases where augmentations are covariant; for example, the task-oriented dialog example is indeed from a real practical use case. This use case doesn’t have anything to do with contrastive learning. Additionally, we are not aware of any work that uses contrastive learning in a covariant data augmentation setting. Hence, we disagree that DAIR needs to be compared head-to-head against contrastive learning for it to be deemed useful. We believe that the consistency regularization for covariant data augmentation is sufficient justification for DAIR to be useful. Finally, we have stated in the broader impact section that the applicability of DAIR beyond the supervised learning setting for which it is proposed/tested remains to be seen.
>
> Based on your feedback, and also feedback from other reviewers, and these clarification points we are working on revising the paper to make the pitch more clear throughout the paper. Having said that, please let us know if you have any other major concerns/comments so that we can address them as we prepare a revision and point-by-point response.
>
> Thanks,\
> Authors

---

> ### Author Response · Authors · 2022-09-18
> **Response to Reviewer mkiw (Part 2 / 2)**
>
> > More analysis on why DAIR is better than feature-level consistency methods, especially, Contrastive learning.
>
> We believe that we have already produced sufficient evidence to support the claim that DAIR is on a par with feature-level consistency regularization on invariant data augmentation but can be also applied to covariant data augmentation to which feature-level consistency regularization is not applicable.
>
> > Numerical experiments of comparing DAIR to CL methods, such as MOCO for example, on ImageNet. Consider using ResNet-50 and ViT as backbone.
>
> We believe this task is out of the scope of this paper, especially because the nature of augmentations used in CL are usually invariant, which is not the main selling point of DAIR.
> In the ImageNet-9 background challenge in Section 4.5 (Previously Section 3.2.3), we have already used ResNet-50 as the backbone, where we show that DAIR remains on a par with KL feature consistency while significantly outperforming ERM and DA-ERM.
>
> > NLP experiments for DIAR v.s. CL are suggested as well. However, I am not an NLP expert so cannot give detailed suggestions.
>
> We compared SimpleTOD + DAIR with SimpleTOD + (Consistency Regularization at feature level ) KL in Table 2 (Previously Table 1), where we show that KL feature consistency hurts the performance. While one can imagine that covariant data augmentations might be used as *negatives* within contrastive learning that begs the question of how to create positives (which probably requires a different invariant data augmentation). In the task-oriented dialog modeling task where the augmentation function creates a dialog context with a different named entity (Figure 2), we can potentially use that as a negative example and there is no clear way to create a positive example for contrastive learning to compare against.
>
>
>
>
>
>
> We again thank the reviewers for their feedback. We hope you find our response satisfactorily addressing your comments. Please let us know if you have any further questions and comments.

---

> > ### Comment · Reviewer_mkiw · 2022-09-18
> > **Thanks for the authors' feedback**
> >
> > Thanks for the authors' feedback. It would be nice to further improve the following parts of this work:
> >
> > * DAIR is designed for covariant data augmentation (DA) that can be applied to many problems that CL cannot. Mix-up is a good example of  covariant DA. However, in most CV problems, covariant DA can be decomposed into variant part + invariant part therefore CL can be applied in the invariant part. For example, all CL methods can be combined with mix-up without difficulty. Can authors find an application that, DAIR is the only choice when using covariant DA. And in this application, using covariant DA s better than using invariant DA.
> >
> > *  While it is shown that DAIR recovers the ground-truth in certain toy examples, it is also possible to show that DAIR cannot recover the ground-truth or is sub-optimal under certain settings. So it would be important to empirically verify whether using DAIR can give better results in real-world tasks.
> >
> > * Lacking large-scale full ImageNet experiments make this work much less convincing. It would be better to conduct these experiments under "standard" setting as previous CL works and analyze / visualize the differences and improvements.

---

> ### Author Response · Authors · 2022-09-18
> **Response to Reviewer mkiw (Part 1 / 2)**
>
> Thank you for your detailed review and we are happy that you believe all contents are well organized and well presented; the toy example given in Sec 2.2 is interesting and insightful; experiments on multiple datasets are with good diversity. We now answer your questions point by point as the following:
>
>
> > One major concern is how to compare DAIR to Contrastive Learning (CL). The authors claimed that the loss-level consistency (as in DAIR) is better than feature-level consistency (as in CL). However, this key argument is not well supported by the theoretical analysis and the numerical experiments:
> > * From theoretical side, it is unclear why DAIR is better than the popular CL approach. The toy example in Sec 2.2 did not show that CL-based method is also guarantee to fail.
> > * From empirical side, there is no experiment to compare DAIR to CL on large-scale dataset and on large models. For example, it is better to compare DAIR to MOCO on ImageNet-1k.
>
>
> We thank the reviewer for bringing up contrastive learning. However, although data augmentation is a critical component of contrastive learning, contrastive learning is not the only use case for data augmentation. The reviewer mentioned MOCO on ImageNet-1k but we believe this task is out of the main focus of the paper as the nature of the augmentations are not covariant. While one can imagine that covariant data augmentations might be used as *negatives* within contrastive learning that begs the question of how to create positives (which probably requires a different invariant data augmentation). For example in the task-oriented dialog modeling task where the augmentation function creates a dialog context with a different named entity (Figure 2), we can potentially use that as a negative example and there is no clear way to create a positive example for contrastive learning.
>
> Additionally, we believe that we have already produced sufficient evidence to support the claim that DAIR is on a par with feature-level consistency regularization on invariant data augmentation but is the only form of consistency regularization that can be also applied to covariant data augmentation to which feature-level consistency regularization is not applicable (See Toy Example 6&9 in Section 2.1, and Section 3).
>
>
>
> > It is unclear what Prop. 4 indicates. For example, how can we choose the $\lambda$ in practice according to Prop. 4?
>
> Prop. 4 connects the discrepancy between the values of $\lambda$ for a perfect invariant solution and the practical solution obtained by DAIR. It gives us an idea on the range of $\lambda$ that needs to be tuned. For example, assuming the cross-entropy loss value of $10^{-2}$ is acceptable in a 10-class classification task, then immediately by Prop. 4 we know $\lambda<=\log(10)/10^{-2}=230$. We agree that the optimal $\lambda$ still needs to be tuned on the validation set and can not be directly obtained by Prop. 4.
>
> > Theorem 5 seems trivial. It follows the classical online learning bound for excess risk. Can authors explain why Theorem 5 is emphasized here?
>
> We disagree. The purpose of Theorem 5 is to theoretically claim that adding DAIR regularizer does not significantly slow down the training process. We show that adding DAIR only introduces a linear penalty on the convergence rate. We also argue that Theorem 5 is not trivial due to the fact not all regularizers have bounded Lipschitz constant of the gradient. With that being said, Theorem 5 helps us understand the cost of adding DAIR regularizer from an optimization perspective.

---

### Review · Reviewer_cKSA · 2022-08-29

**Summary Of Contributions:**

This paper proposes a simple and versatile consistency regularization method called DAIR (data augmented invariant regularization). While conventional consistency regularization evaluates consistency at the intermediate feature level, the proposed method evaluates consistency at the loss level. Therefore, conventional consistency regularization methods can only be applied to invariant data augmentations, but the proposed method is applicable for both invariant and covariant data augmentations. In particular, this paper focuses on DAIR-SQ ($(\sqrt{l(z;\theta)} - \sqrt{l(z;\theta)} )^2$), a special form of DAIR, and provides some theoretical proof of this property. Furthermore, DAIR is experimentally evaluated for various problems ranging from robust classification against adversarial attacks to domain generalization in the presence of environmental shifts. It is shown to achieve comparable or better performance than empirical risk minimization plus data augmentation (DA-ERM) and conventional consistency regularizations.

**Broader Impact Concerns:**

There are no broader impact concerns.


**Requested Changes:**

The reviewer enjoyed reading this well-written paper. However, to better understand this research, please answer the following?

- In Sec. 2.4, this paper states, "At the beginning of training when the network is not yet trained, the loss values on the original samples are large, and $\mathcal{R}_{sq}$ regularizer is weak letting the training to proceed towards a good solution for the original samples." If this is true, can we use DAIR-L1 to obtain similar results by using a smaller value of $\lambda$ in the early stages of learning and increasing the value of $\lambda$ as the learning progresses?

- Is it possible to evaluate the effectiveness of the proposed method as the strength of correlation between inputs and outputs changes? For example, in Colored MNIST, the effectiveness of DAIR-SQ  will be measured according to changes in the probability of color since the strength of the proposed method is that DAIR-SQ is applicable to covariate data augmentation.

- In the experiment on the real dataset, the value of $\lambda$ is determined by grid search on the validation data. Is it possible to find an appropriate value of $\lambda$ on the colored MNIST dataset described in section 2.3? The reviewer is interested to see the difference between the optimal value in Fig.6 and the ones from grid search. In the realistic case, the distribution of validation data is assumed to be the same as the training data.

- In the case of neural task-oriented dialog modeling, SimpleTOD is used, and in VQA, SAAA is used as the base model. Can we get the same effect by using other base models?


**Strengths And Weaknesses:**

Strengths
- The paper is well written, and the proposed method is easily understood.
- This paper performed thorough experiments on various problems ranging from robust classification against adversarial attacks to domain generalization in the presence of environmental shifts.
- This paper conducted a theoretical analysis of the optimality of DAIR-SQ versus DA-ERM in linear regression and theoretical analysis of the convergence of DAIR-SQ.
- While conventional consistency regularization methods can only be applied to invariant data augmentations, DAIR-SQ can be used for both invariant and covariant data augmentations.


Weaknesses
- Theoretical analysis and experiments in this paper focus only on DAIR-SQ, a special form of DAIR, and do not cover DAIR in general.
- The proposed method requires pairing information of the original data and its augmented data.
- The current method for determining the hyperparameter $\lambda$ is ad hoc, such as grid search.
- Theoretically, the value of the hyperparameter $\lambda$ can be set to infinity to obtain the optimal test error, while convergence becomes linearly worse as the value of $\lambda$ increases.

---

> ### Author Response · Authors · 2022-09-18
> **Tables used in our response**
>
> **ResponseTable 1: Tuning $\lambda$ with different validation schemes. *Random Augmentation* is used here.**
>
> | $\lambda$ | Test ($p=0.9$) | Val on Aug (Rnd) | Val on Train ($p=0.2$) | Val on DA | Val on $p = 0.9$ (Oracle) |
> |:---------:|:--------------:|:----------------:|:----------------------:|:---------:|:-------------:|
> |    1.00   |      62.46     |       69.13      |          73.05         |   71.09   |     62.65     |
> |    1.58   |      68.34     |       70.67      |          73.11         |   71.89   |     68.48     |
> |    2.51   |      68.96     |       70.35      |          72.63         |   71.49   |     68.98     |
> |    3.98   |      69.13     |       69.64      |          71.44         |   70.54   |     68.94     |
> |    6.31   |      71.08     |       70.85      |          71.61         |   71.23   |     71.16     |
> |   10.00   |      71.74     |       71.81      |          72.16         |   71.98   |     72.12     |
> |   15.85   |      71.06     |       70.75      |          71.31         |   71.03   |     71.22     |
> |   25.12   |      73.24     |       72.76      |          73.20         |   72.98   |     73.16     |
> |   39.81   |      72.75     |       72.83      |          73.05         |   72.94   |     73.30     |
> |   63.10   |      72.82     |       72.30      |          72.97         |   72.63   |     72.95     |
> |   100.00  |      73.07     |       72.86      |          73.12         |   72.99   |     73.15     |
> |   158.49  |      73.12     |     **72.92**    |        **73.35**       | **73.14** |   **73.41**   |
> |   251.19  |    **73.37**   |       72.90      |          73.24         |   73.07   |     73.21     |
> |   398.11  |      72.18     |       71.90      |          72.46         |   72.18   |     72.23     |
> |   630.96  |      72.45     |       72.32      |          72.69         |   72.50   |     72.44     |
> |  1000.00  |      71.21     |       70.85      |          71.44         |   71.14   |     71.29     |
> |  1584.89  |      70.92     |       70.68      |          71.26         |   70.97   |     71.08     |
> |  2511.89  |      69.96     |       69.83      |          70.58         |   70.20   |     70.16     |
> |  3981.07  |      51.15     |       50.37      |          50.58         |   50.48   |     49.91     |
> |  6309.57  |      63.50     |       63.79      |          64.27         |   64.03   |     64.35     |
> |  10000.00 |      48.85     |       49.63      |          49.42         |   49.53   |     50.09     |
>
> **ResponseTable 2: Tuning $\lambda$ with different validation schemes. *Adversarial Augmentation* is used here.**
>
> | $\lambda$ | Test ($p=0.9$) | Val on Train ($p=0.2$) | Val on DA | Val on $p=0.3$ | Val on $p=0.4$ | Val on $p = 0.9$ (Oracle) |
> |:---:|:---:|:---:|:---:|:---:|:---:|:---:|
> | 1.00 | 47.00 | 74.39 | 76.80 | 70.66 | 66.93 | 47.11 |
> | 1.58 | 51.04 | 74.81 | **77.02** | 71.98 | 68.30 | 50.20 |
> | 2.51 | 51.13 | **75.02** | 76.98 | 71.80 | 68.32 | 50.36 |
> | 3.98 | 53.64 | 74.71 | 76.57 | 72.09 | 68.50 | 53.61 |
> | 6.31 | 53.58 | 74.92 | 76.79 | 72.02 | 68.92 | 53.53 |
> | 10.00 | 60.30 | 74.49 | 75.66 | 72.43 | 70.71 | 59.97 |
> | 15.85 | 60.64 | 74.35 | 75.83 | 72.45 | 70.54 | 59.76 |
> | 25.12 | 67.54 | 73.67 | 74.46 | **73.24** | 72.20 | 67.35 |
> | 39.81 | 69.00 | 73.33 | 73.93 | 72.42 | 72.06 | 68.46 |
> | 63.10 | 69.90 | 72.96 | 73.54 | 72.87 | 72.11 | 69.52 |
> | 100.00 | 72.15 | 71.86 | 72.29 | 72.24 | 72.08 | 71.59 |
> | 158.49 | 71.59 | 70.93 | 71.26 | 71.37 | 71.20 | 71.09 |
> | 251.19 | 71.98 | 71.97 | 72.10 | 72.13 | 71.40 | 71.89 |
> | 398.11 | 73.28 | 72.37 | 72.66 | 72.79 | 72.59 | 72.62 |
> | 630.96 | 73.16 | 72.55 | 72.79 | 72.58 | 72.53 | 72.73 |
> | 1000.00 | 73.26 | 72.74 | 72.95 | 72.98 | **72.83** | **72.90** |
> | 1584.89 | 73.06 | 72.59 | 72.78 | 72.73 | 72.51 | 72.51 |
> | 2511.89 | 72.84 | 72.29 | 72.45 | 72.69 | 72.45 | 72.44 |
> | 3981.07 | **73.28** | 72.52 | 72.87 | 73.01 | 72.75 | 72.79 |
> | 6309.57 | 72.25 | 71.93 | 72.15 | 72.01 | 71.90 | 72.10 |
> | 10000.00 | 73.10 | 72.42 | 72.54 | 72.92 | 72.44 | 72.52 |

---

> ### Author Response · Authors · 2022-09-18
> **Response to Reviewer cKSA (Part 2 / 2)**
>
> > In the experiment on the real dataset, the value of $\lambda$ is determined by grid search on the validation data. Is it possible to find an appropriate value of $\lambda$ on the colored MNIST dataset described in section 2.3? The reviewer is interested to see the difference between the optimal value in Fig.6 and the ones from grid search. In the realistic case, the distribution of validation data is assumed to be the same as the training data.
>
> We thank the reviewer for pointing this out. To answer your question, we repeated the experiment in Figure 9 (Previously Figure 6) with multiple validation schemes. ResponseTable 1 and ResponseTable 2 above show the test accuracy and the validation accuracy with different validation schemes. We marked the best validation accuracy for each scheme (in each column) as **bold**. For the *Random Augmentation*, in ResponseTable 1, the validation scheme seems to have little impact on the selected $\lambda$ and the test performance. Tuning $\lambda$ on the validation set with either training transformation scheme or DA (the average of training scheme validation accuracy and augmentation scheme validation accuracy) results in similar good results as Oracle tuning where validation data follows the same distribution as test data. For the *Adversarial Augmentation*, in ResponseTable 2, we see that tuning the validation with the training validation scheme (or also DA validation scheme) results in a significant performance drop compared to Oracle validation while still outperforming DA-ERM. The reason is that, by design the *Adversarial Augmentation* scheme is difficult. Recall that $p$ in *Adversarial Augmentation* determines the correlation of the label with color (see Appendix D). Hence, the augmentation scheme ($p=0.1$) is further away from the testing scheme ($p=0.9$) compared with the training scheme ($p=0.2$). Subsequently,  $\lambda$ chosen on the DA scheme is not expected to work well at test time due the nature of the distribution shift. Moreover, we see that as validation $p$ gets closer to the test scheme ($p=0.9$),   we obtain models with more favorable test performance which is expected. In particular,tuning with ($p=0.4$) results in the same optimal $\lambda$ as the one we get in tuning with the Oracle validation set ($p=0.9$).
>
>
>
>
> > In the case of neural task-oriented dialog modeling, SimpleTOD is used, and in VQA, SAAA is used as the base model. Can we get the same effect by using other base models?
>
>
> We have no reason to believe that the benefits of DAIR are tied to these network structures, as we have observed in numerous experiments with various backbone networks. To address the reviewer’s comment, we are running the task-oriented dialog experiment on both GPT-2 and BART backbones for comparison. However, due to the complexity of the task itself, the experiment results are not available at this point yet. We will follow up here right after we obtain the results.
>
> We again thank the reviewers for their constructive feedback. Please let us know if you have any further questions and concerns.

---

> ### Author Response · Authors · 2022-09-18
> **Response to Reviewer cKSA (Part 1 / 2)**
>
> Thank you for your detailed review and we are pleased that you enjoyed reading our paper. We answer your questions point by point as the following:
>
> > Theoretical analysis and experiments in this paper focus only on DAIR-SQ, a special form of DAIR, and do not cover DAIR in general.
>
> Yes, we agree with the reviewer that the theoretical analysis only applies to DAIR-SQ. However, as we have observed on various experiments DAIR-SQ was the only variant with promising practical performance that warranted a formal theoretical investigation. For example, we empirically show DAIR-SQ significantly outperforms DAIR-L1 in several applications in Section 4.1 and 4.2 (Previously Section 2.2 and 2.3).
>
> > The proposed method requires pairing information of the original data and its augmented data.
>
> Yes, that is correct and we have acknowledged this point as one of the limitations in Section 6 on page 13 (Previously  Section 5 on page 12). However, we note that counterfactual data augmentation is widely used in practice, which creates such pairing information for free. Given the abundance of such methods in the literature, DAIR could potentially be added on top of them with marginal cost and boosts the performance. In this sense, we consider DAIR to be still broadly applicable even in light of this requirement. In particular, we did not devise any of the augmentation functions used in the numerical experiments in Section 3 and Section 4 (Previously Section 3), all of which already existed in the literature.
>
> > The current method for determining the hyperparameter is ad hoc, such as grid search.
>
> Yes, we agree with the reviewer that the hyperparameter ($\lambda$) needs to be tuned.  Having said that, this is very common and standard for algorithms in deep learning. In particular, every other consistency regularization technique that we are aware of suffers from the same requirement.
>
> > Theoretically, the value of the hyperparameter can be set to infinity to obtain the optimal test error, while convergence becomes linearly worse as the value of $\lambda$ increases.
>
> In light of a perfect solver with no run-time constraint the higher value of $\lambda$ is preferrable. However, given a training budget in Theorem 4 on page 7 (Previously Theorem 5 on page 8), we theoretically showed that the value of $\lambda$ directly hurts the number of steps needed for convergence. We also empirically confirmed this issue in Section F.2 (see Table 17). Similar behavior has been observed for traditional nonconvex regularizers such as SCAD in the past.
>
>
>
> > In Sec. 2.4, this paper states, "At the beginning of training when the network is not yet trained, the loss values on the original samples are large, and $\mathcal{R}_\text{sq}$ regularizer is weak letting the training to proceed towards a good solution for the original samples." If this is true, can we use DAIR-L1 to obtain similar results by using a smaller value of $\lambda$ in the early stages of learning and increasing the value of $\lambda$ as the learning progresses?
>
> This is complicated. We tried to devise  a way of scheduling $\lambda$ such that it increases gradually as training proceeds to make the optimization procedure easier.  However, we haven’t found a working schedule (even though admittedly we only tried a few tricks only). To make our argument more convincing, we did a sanity check in which we started with the DAIR-SQ solution and continued training with the L1 regularizer and a large $\lambda$, where it turned out that L1 does not hurt and the training did not diverge. Clearly this sanity check shows that L1 regularizer itself is valid in terms of optimization landscape but it is difficult for L1 to lead the model to converge to a good solution. This observation indeed shows the superiority of DAIR-SQ where we can just pick a fixed $\lambda$ without adding any other ad hoc scheduling.
>
> > Is it possible to evaluate the effectiveness of the proposed method as the strength of correlation between inputs and outputs changes? For example, in Colored MNIST, the effectiveness of DAIR-SQ will be measured according to changes in the probability of color since the strength of the proposed method is that DAIR-SQ is applicable to covariate data augmentation.
>
> We thank the reviewer for this suggestion. In the new toy example (Rotated 6&9 in Section 2.1), we create different degrees of distribution shift between train and test to measure how effective different methods are as we scale such parameters. We hope the reviewer will find the new experiments insightful where we show that DAIR-SQ is the only method which consistently provides gains over ERM with such covariant data augmentation scheme.

---

### Author Response · Authors · 2022-09-18
**Response to All Reviewers & Summary of Revisions**

We thank all the reviewers for their thorough and constructive feedback. We are pleased that you find our paper “well written”, “easy to read”, “well organized and well presented”. We are also happy that you like our theoretical results. We also received actionable feedback to improve the presentation, and revised our paper accordingly. Here we summarize the main modifications of our paper:
*  To motivate DAIR, especially as it relates to covariant data augmentation, we introduce a new toy example in Section 2, namely Rotated 6 & 9. This example is based on covariant data augmentation demonstrating the importance of enforcing consistency at loss level. It also shows that consistency regularization at feature level (such as KL feature consistency) in covariant applications hurts the performance. We hope this change will answer the comment from **reviewer mkiw**.
*  The original toy example in Section 2.2 is relegated to Appendix A. As pointed out by the reviewers, the toy example was not completely aligned with the flow of the paper. The reason we moved it to the appendix (instead of completely removing it from the paper) is that some of the reviewers found the example and the related theoretical results insightful, especially to showcase that the benefits of DAIR are complementary to regularization.
*  We adopted the suggestion from **reviewer TWyi** on the organization of the paper.  We reorganized Section 2 to only contain motivating toy examples and theoretical results. We moved the experiments from Section 2.3 to section 4.1 and 4.2. Original sections 2.4, 2.5 and 2.6 which discuss the properties of DAIR are merged together.
* Original section 3.1 (covariant experiments) and section 3.2 (invariant experiments) are re-labeled as section 3 and section 4 for better readability.

Please let us know if you have any further comments/questions and we are happy to respond to them. Thanks again for your valuable feedback; we hope you find our revisions satisfactorily addressing your comments/questions.

Sincerely, \
Authors

---

### Decision · Action_Editors · 2022-10-13

**Recommendation:** Accept with minor revision

**Comment:**

This paper addresses the issues that appear when applying regularization via data augmentation to problems in which enforcing the labels of augmented examples being same seems less meaningful, as the motivating examples of covariant tasks shown in Fig.1 and Fig. 2 in the paper. To address the issue, the paper argues for data-augmented invariant regularization (DAIR) at the LOSS level, rather than the conventional feature consistency regularization.

The paper presents a few algorithm variants of DAIR at the loss level, including DAIR-SQ and DAIR-L1 (Garg et al., 2019), and further shows that DAIR-SQ lower bounds DAIR-L1 (Lemma 1 in the paper), leading to better regularization solutions when training with DAIR-SQ as the regularizer. The paper further analyzes how the penalty parameter \lambda of the DAIR-SQ regularizer affects performance by showing that DAIR-SQ leads to convergent algorithms when optimized by gradient descent, where classification with logistic loss is assumed (Proposition 3 and Theorem 4 in the paper).

The paper presents experiments of both covariant and invariant tasks; results confirm the effectiveness of DAIR at the loss level.

The paper receives mixed reviews from 4 reviewers. While all of them admit some merits of the proposed issue of DAIR at the loss level, they point out main concerns such as the lack of comparisons with contrastive learning, no use of pairs of samples both of which are augmented ones, the lack of large-scale experiments (e.g., on the ImageNet), etc.

The biggest concern is, however, how the paper can be positioned in the literature. The paper claims that regularization at loss level has been relatively unexplored, however, reviewer P15o points out “consistency regularization no matter at feature level nor loss level has been widely used in different fields, such as semi-supervised learning, knowledge distillation and adversarial training [1][2][3]. There is no essence different from DAIR with the existing works.”

AE agrees. In fact, the key novelty of the paper is its improvement of DAIR-SQ over DAIR-L1 (Garg et al., 2019, as already discussed in the paper), and also Proposition 3 and Theorem 4 that additionally support the improvement. As such, the paper should be structurally reformed; for example, the paper may first present the rich literature of DAIR at the LOSS level, in particular those techniques (e.g., DAIR-L1, the regularization methods presented in [1], and many others) closely related to the proposed DAIR-SQ, and then discuss how these existing techniques are limited or suboptimal in addressing covariant tasks; the real contribution of this paper about DAIR-SQ and its supporting analyses then follows; as such, DAIR-SQ should be the real method proposed in the paper, rather than DAIR. Please revise the paper following the above suggestions, including other minor issues pointed out by the reviewers.

[1] Virtual Adversarial Training: A Regularization Method for Supervised and Semi-Supervised Learning. TPAMI 2019.


**Audience:**

Yes

**Claims And Evidence:**

The claims are largely supported by theoretical and empirical evidence. However, writing and organization of the paper should be revised.

---

> ### Author Response · Authors · 2023-01-22
> **Response to requested minor revisions**
>
> Dear Kui,
>
> Thank you again for leading the review process of our paper. We are excited that our paper has been accepted with the requested minor revisions. We incorporated your constructive feedback and the comments from the reviewers in our revision. We would also like to clarify our viewpoint on a question/remark raised through the review process and get your feedback on it:
>
> We agree that consistency regularization has been extensively studied in the literature, and we better clarified this in our revision (see Introduction on page 2 and the explanation right after the definition of DAIR on page 3). However, the references raised by the reviewers and the ones we cited all apply regularization at the feature level (e.g. at logits) and not at the loss level. In particular, references [1][2][3] mentioned by reviewer P15o applied consistency regularizations at the logit (feature) level. This is discussed further in our revision (see the explanation right after the definition of DAIR in section 2 on page 3 and footnote on page 4). We also discussed in section 2.1 why feature-level regularization cannot solve covariant problems (unlike our proposed DAIR loss level regularization). Finally, L1 regularization used in Garg et al., 2019 is also at the logit (feature) level. We added a short discussion explicitly around loss-level v.s. feature-level regularization when we introduce DAIR in our revision (see the explanation right after the definition of DAIR in section 2 on page 3). We also added a short discussion on L1 regularization used by Garg et al., 2019 to clarify the above misunderstandings (see footnote on page 4).
>
> Lastly, we have also updated the results of VQA and separated Invariant VQA (now Section 4.3) and Covariant VQA (now Section 3.2) results to better showcase the applicability of DAIR vs other regularizers to these problems.  and released our code here: https://github.com/optimization-for-data-driven-science/DAIR.
>
> Please let us know if these modifications addressed your concerns.
>
> Thank you again, and we look forward to hearing back from you.
>
> Sincerely, \
> Authors